# Application of Laser Welding in Electric Vehicle Battery Manufacturing: A Review

**Junbo Feng** [1,2], **Peilei Zhang** [1,2,*] , **Hua Yan** [1,2], **Haichuan Shi** [1,2], **Qinghua Lu** [1,2], **Zhenyu Liu** [1,2], **Di Wu** [1,2], **Tianzhu Sun** [3], **Ruifeng Li** [4] and **Qingzhao Wang** [5]

[1] School of Materials Science and Engineering, Shanghai University of Engineering Science, Shanghai 201620, China; 18990122132@163.com (J.F.); yanhua@foxmail.com (H.Y.); shc0010@126.com (H.S.); luqh@sues.edu.cn (Q.L.); zhenyu_ender@foxmail.com (Z.L.); wudi612@126.com (D.W.)
[2] Shanghai Collaborative Innovation Center of Laser Advanced Manufacturing Technology, Shanghai 201620, China
[3] Warwick Manufacturing Group (WMG), University of Warwick, Coventry CV4 7AL, UK; tianzhu.sun@warwick.ac.uk
[4] School of Materials Science and Engineering, Jiangsu University of Science and Technology, Zhenjiang 212003, China; li_ruifeng@just.edu.cn
[5] SANY Heavy Industry Co., Ltd., Shanghai 201306, China; wangrain9_14@163.com
* Correspondence: peilei@sues.edu.cn

**Abstract:** Electric vehicle battery systems are made up of a variety of different materials, each battery system contains hundreds of batteries. There are many parts that need to be connected in the battery system, and welding is often the most effective and reliable connection method. Laser welding has the advantages of non-contact, high energy density, accurate heat input control, and easy automation, which is considered to be the ideal choice for electric vehicle battery manufacturing. However, the metal materials used for the electrodes of the battery and the connectors used to connect the battery are not the same, so the different materials need to be welded together effectively. Welding different materials together is associated with various difficulties and challenges, as more intermetallic compounds are formed, some of which can affect the microstructure, electrical and thermal properties of the joint. Because the common material of the battery housing is steel and aluminum and other refractory metals, it will also face various problems. In this paper reviews, the challenges and the latest progress of laser welding between different materials of battery busbar and battery pole and between the same materials of battery housing are reviewed. The microstructure, metallographic defects and mechanical properties of the joint are discussed.

**Keywords:** electric vehicle battery; laser welding; welding defects; dissimilar metal; identical metal

## 1. Introduction

Due to global warming, today's climate problems are intensifying, and extreme weather is occurring frequently. The main cause of this problem group is greenhouse gas emissions, mainly carbon dioxide (90%), and the transport sector is one of the largest contributors to greenhouse gas emissions, according to the International Energy Agency (IEA), and in 2015, global $CO_2$ emissions reached 323 billion tons, while transport accounted for 24% of the total emissions. Three quarters of this is contributed by the road component [1]. To mitigate climate change, carbon emission laws have been enacted around the world [2,3]. New energy vehicles (NEV), as an alternative to traditional internal combustion engine vehicles (ICEV), are rapidly developing in major international automotive markets. China, the United States, Japan, Germany and other countries have restricted the sales of traditional internal combustion engine vehicles at the national level and formulated a series of new policies to encourage the development of new energy vehicles, so as to reduce the use of oil and reduce carbon dioxide emissions [4].

The fastest developing new energy vehicles are electric vehicles (EVs), which are powered by power batteries. Lithium-ion battery has become the most important power supply for electric vehicles because of its high energy density, low self-discharge and long life cycle [5–7]. Batteries used in electric vehicles are mainly small solid cylindrical batteries, large solid prismatic batteries and large soft bag or polymer batteries [8–10], as shown in Figure 1. Battery packs for electric vehicles are usually designed and manufactured in a battery-module-cell structure, as shown in Figure 2. The main difference in practice is how to achieve the required battery capacity and power. A small number of large capacity cells can be connected in series, as shown in Figure 2a. Alternatively, multiple small batteries with small capacity are connected in parallel and then connected in series to form high-capacity modules, as shown in Figure 2b. These batteries are usually connected by busbars [11], as shown in Figure 3. Power batteries usually work in harsh driving environments, such as vibration, high temperature and possible collision. How to securely connect hundreds of connections in battery modules is related to the new performance and safety of the entire battery system. Various bonding techniques, such as laser welding, friction stir welding, tungsten inert gas welding, ultrasonic lead bonding and resistance spot welding, have been used in battery manufacturing [8,10,12]. Ultrasonic welding mainly uses high-frequency vibration, usually 20 kHz or above, to connect materials by forming solid-state bonds under clamping pressure [13]. Ultrasonic welding is suitable for the welding of multiple thin foils, dissimilar materials or highly conductive materials. It is mainly used in banded batteries [14], and electric vehicle batteries are usually cylindrical or prismatic batteries, which may destroy the integrity of the battery structure under the action of pressure and vibration, so it is not suitable for the welding of electric vehicle batteries [8]. The working principle of resistance spot welding is mainly to apply pressure on the contact surface of the workpiece and connect large current to cause partial melting of the workpiece [12]. However, the commonly used materials in electric vehicle batteries are aluminum and copper, and aluminum and copper have the characteristics of high electrical and thermal conductivity, so resistance welding is difficult to weld. Laser welding is considered to be the most promising connection method because of its easy automation, high accuracy, small heat-affected zone, non-contact process, high process speed and ease of welding different metals. Laser welding is an efficient and precise welding method using high energy density laser beam as heat source. Due to heat concentration, fast welding speed, small thermal effect, small welding deformation, easy to realize efficient automation and integration [15–17], it is more and more widely used in power battery manufacturing.

| Cylindrical cell | Prismatic cell | Pouch cell |
|---|---|---|
| • Small size (e.g. 18650 type (ø 18 mm, height 650 mm)) <br> • Hard casing <br> • Low individual cell capacity <br> • Build in safety features <br> • Comparably cheap | • Hard casing <br> • Large size <br> • High individual cell capacity | • Soft casing <br> • Large size <br> • High individual cell capacity <br> • Geometrical deformation during (dis-)charging |

**Figure 1.** Overview of different cell types used in automotive battery applications: (**left**) cylindrical cell, (**middle**) prismatic cell, and (**right**) pouch cell. Reprinted with permission from Ref. [10]. Copyright 2020, Elsevier.

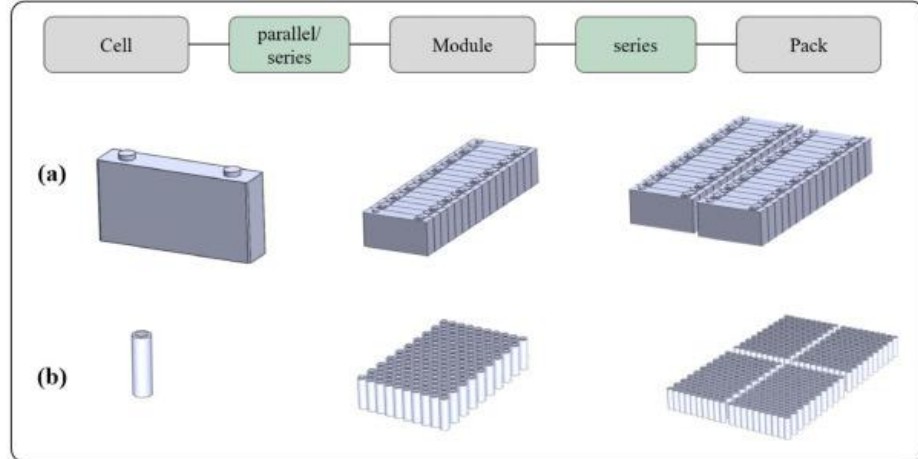

**Figure 2.** Overview of battery packs indicating two constructions with (**a**) cylindrical and (**b**) prismatic cells. Reprinted with permission from Ref. [10]. Copyright 2020, Elsevier.

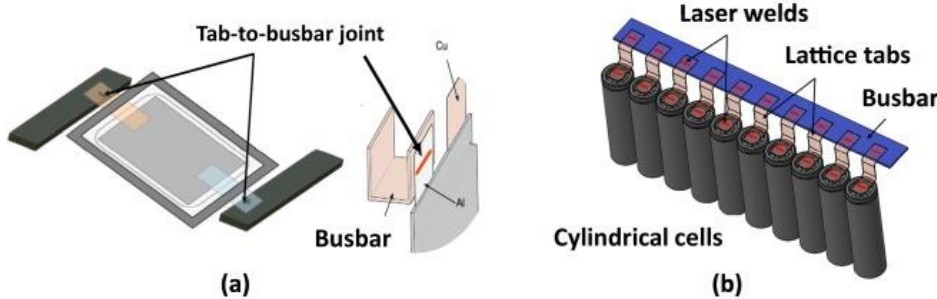

**Figure 3.** An illustration of tab-to-busbar joints made during (**a**) pouch cell-based and (**b**) cylindrical cell-based module manufacture [11].

In addition, the battery connection can be mechanically fastened in a variety of ways, including nut bolts, spring fasteners, screws or fasteners [18]. Nut and bolt joints may be either physically distinct nut and bolt assemblies or a threaded feature, for example, electrode and nut. For battery module level connections, nut and bolt joints are mainly limited to prismatic cells. Some special battery modules are not suitable for permanent connection (such as welding) due to the need for battery maintenance, so mechanical nuts and bolts can also be used for connection of special battery systems. At present, battery casings are mainly produced by welding sheets, so some welding defects, such as pores and cracks, are inevitable. Niu et al. [19] added high entropy alloy to aluminum powder during additive manufacturing of aluminum, which inhibited the crack generation and improved the strength. Therefore, with the development of laser additive, the battery case may be produced by additive manufacturing in the future.

This paper mainly reviews the laser welding of dissimilar metal joints between battery and bus in electric vehicle battery system, as well as the packaging of the same metal between battery pack by laser welding. The difficulties and challenges of laser welding between homogenous and dissimilar metals are discussed. Especially, the welding defects caused by the welding between dissimilar metals, the causes of these welding defects and the possible solutions are put forward.

## 2. Welding between Batteries and Busbars

In all the production processes of power battery packs, there is a key process, that is, the welding of a single lithium battery and the connector. This is the key to the quality of series and parallel lithium-ion battery cells, that is, the welding of the battery pole and the busbars. The quality of the welding here will directly affect the reliability of the quality of the lithium-ion battery pack used as a power source for electric vehicles. In addition, due to

the relative particularity of lithium-ion battery, the welding technology has also put forward high requirements. If the welding strength is weak, the internal resistance of the battery string will increase, thus affecting the normal power supply of the battery string. Excessive welding heat will cause the electrode cover of the battery core to be penetrated, resulting in electrolyte leakage and battery circuit's short circuit, resulting in battery combustion or even explosion, which seriously threatens the safety of passengers and drivers. It is because of the problems of unreliable welding quality and low welding efficiency in series and parallel welding of power battery pack that the safety and production efficiency of power battery pack are very low. Therefore, in order to ensure the safety of its use and production efficiency, the welding between the battery pole and the busbars must be reliable, which is an important factor to ensure the product yield and service life.

### 2.1. Aluminum and Steel

Battery busbars are made of two common materials: copper and aluminum. The battery electrode materials are usually steel and aluminum, and the parameters and challenges of laser welding are different. Aluminum has the advantages of good electrical conductivity, light weight and good plasticity, so it is very suitable as a busbar material. The efficient and reliable connection of steel and aluminum can provide huge economic benefits for battery manufacturing.

However, the connection between the aluminum busbars and the steel poles of the battery is challenging because iron and aluminum have great differences in thermal physical properties such as melting point, thermal conductivity and thermal expansion coefficient. In addition, the low solubility between Fe and Al leads to the formation of brittle intermetallic layers where iron and aluminum are metallurgically incompatible, and the resulting fusion welding is prone to the formation of harmful intermetallic compounds (IMCs). The formation of intermetallic compounds (IMCs) has been shown to cause a variety of welding defects, such as microcracks and pores [20–24]. The chemical composition, crystal structure, hardness and Gibbs free energy of various IMCs formed in Fe–Al binary system are shown in Table 1 [25–27]. $Fe_2Al_5$, $Fe_4Al_{13}$ and$FeAl_2$ are Al-rich phases, and FeAl and $Fe_3Al$ are Fe-rich phases. As can be seen from Table 1, aluminum-rich IMCs have stronger hardness and brittleness than iron-rich IMCs, so cracks and other defects are more likely to occur between welded joints. The iron-rich IMCs have better toughness and ductility, which can reduce the generation of cracks. However, in terms of Gibbs free energy, the formation of the Al-rich phase is thermodynamically more favorable to the formation of the Fe-rich phase. $Fe_2Al_5$ is thermodynamically more stable, forming first, followed by $Fe_4Al_{13}$, $FeAl_2$, FeAl, and $FeAl_3$ [28]. IMCs are usually resistive, and too much IMCs will increase the internal resistance of the battery system, resulting in more Joule heat generated during the charging and discharging process of the battery system, affecting the life of the battery system. Therefore, the generation of IMC phase in the weld tissue should be controlled as much as possible during the welding process.

**Table 1.** Fe–Al IMC properties.

| Phases | Al at% | Hardness, HV | Crystal Structure | $\Delta G$ (KJ mol$^{-1}$) |
|---|---|---|---|---|
| $Fe_2Al_5$ | 70–73 | 1000–1100 | orthorhombic | −19.64 |
| $Fe_4Al_{13}/FeAl_3$ | 74.5–76.6 | 820–980 | BC monoclinic | −22.87 |
| FeAl2 | 66–66.9 | 1000–1050 | triclinic | −17.0 |
| FeAl | 23–55 | 400–520 | Simple cubic (B2 type) | −11.09 |
| $Fe_3Al$ | 23–34 | 250–350 | FCC | −4.83 |

Yang et al. [29] found that when welding aluminum and steel, the penetration depth should be controlled within a certain range, and Fe–Al IMCs rich in iron are mainly formed when the penetration depth was low. When the penetration depth was increased, the aluminum rich Fe–Al IMCs were mainly formed, which would make the mechanical properties of the joint worse. Chen et al. [30] added magnetic field action perpendicular to the welding direction when welding 301 stainless steel and 5754 aluminum alloy. They

found that increasing the magnetic field inhibited the diffusion of C atoms and reduced the austenite grain size. Increasing the magnetic field could also effectively inhibit the concentration of Al in the joint, as shown in Figure 4, thus reducing the cracks and thickness of IMCs at the interface, improving the shear strength of the joint and reducing the hardness of the joint.

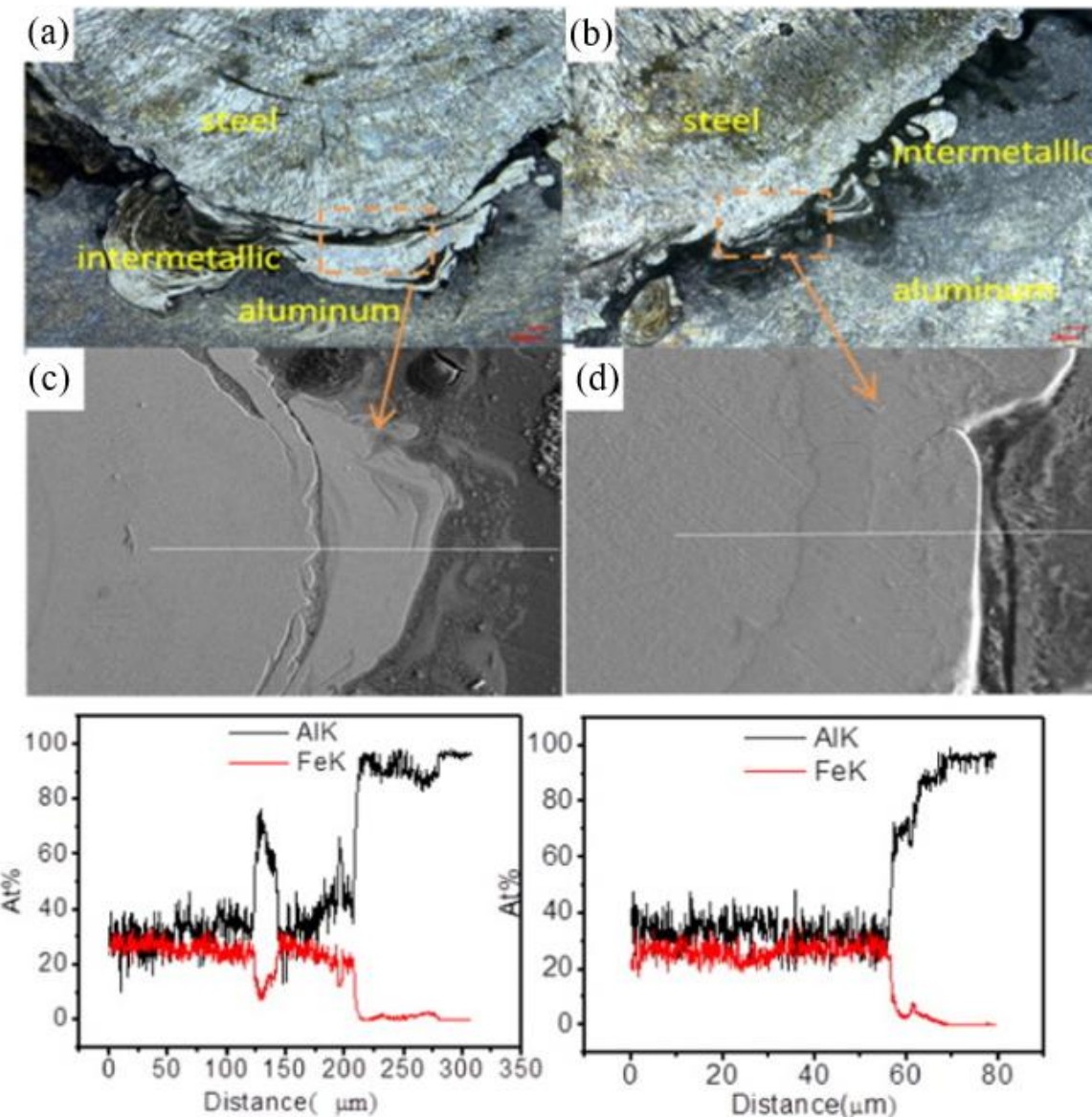

**Figure 4.** Morphology of interface of two groups: (**a**) B = 0 and (**b**) B = 240 mT; Line scanning area and analysis of the seam–aluminum interface layer: (**c**) B = 0 and (**d**) B = 240 mT. Reprinted with permission from Ref. [30]. Copyright 2016, Elsevier.

Torkamany et al. [31] welded 0.8 mm thick mild steel (st14) with a 2 mm thick 5754 aluminum alloy. They found that when the power of the pulsed laser was too high, it was not conducive to the formation of the weld. When the laser power was higher, it increased the mixing of steel and aluminum, increased the content of aluminum in the weld and formed more intermetallic compounds. Increasing the duration of the laser pulse would have a similar effect with the increase in the amount of heat input. These regions with more intermetallic compounds form cracks under thermal stress, as shown in Figure 5. On the other hand, reducing the duration of the laser pulse below a critical level led to a lack of fusion. Increasing the welding speed would also lead to incomplete interface

fusion and reduce joint strength. They reported optimal values of process parameters for producing high-strength welds due to low intermetallic compound content, high surface quality and no obvious defects in the continuous interface layer. The peak power was 1430 W, the pulse duration is 5 ms, and the welding speed was 4 mm/s.

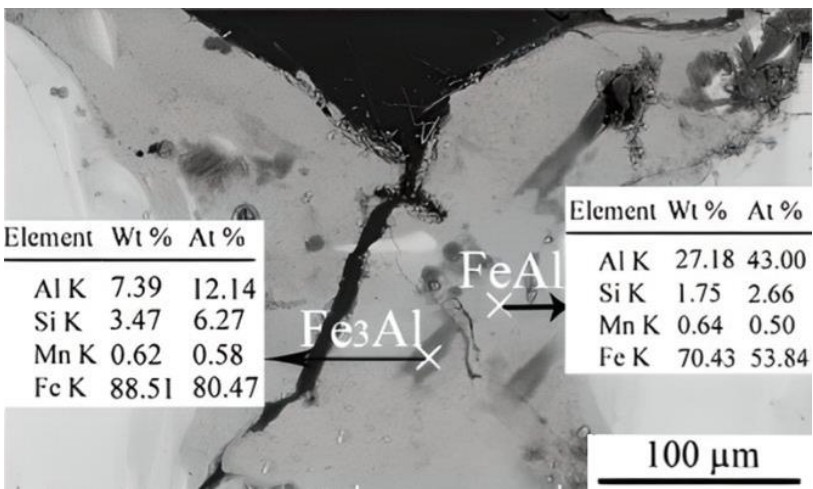

**Figure 5.** SEM micrograph around the joint crack. Reprinted with permission from Ref. [31]. Copyright 2010, Elsevier.

Chen et al. [32] studied the effect of intermediate nickel foil layer on welding of A5052 aluminum alloy and 201 stainless steel. The images with and without nickel layers are shown in Figure 6a,b. In the weld with nickel layer, an intermetallic layer could be obviously seen, which can be divided into $FeAl_3$ and $Al_{0.9}Ni_{1.1}$ layers, as shown in Figure 6c, indicating that nickel foil changed the composition of intermetallic compounds. They believed that because Al had a certain solubility in $\alpha$-Fe, when aluminum was mixed into the molten steel as a solute element, no intermetallic compounds were observed in the fusion zone. The welding depth had a significant effect on the mechanical properties of the weld. Initially, the tensile strength increased as the welding depth reaches 300 $\mu$m, but as the welding depth further increased, the tensile strength began to decrease, which was due to the higher aluminum content, forming a more brittle intermetallic compound. In addition, tensile tests and microhardness measurements of welded samples showed that Ni foil increased the tensile strength while reducing the microhardness of the intermetallic layer.

Cao et al. [33] shifted the focus of laser welding to the stainless steel side when welding aluminum alloy and stainless steel. They found that a certain amount of laser offset could effectively improve the tensile strength of the weld. When the laser offset was 0.2 mm, the tensile strength of the welded joint reaches 129.6 MPa. Wei et al. [34] found that when welding SUS3010S stainless steel and 6061 aluminum alloy, laser cleaning of the stainless steel layer could effectively enhance the bonding strength of the welded joint. They used the laser to clean the stainless steel at the same time, so that the stainless steel surface under the action of the laser to form a fish scale pit. They found that the molten Al could spread and fill the scale pits on these stainless steel surfaces, resulting in a significant increase in the mechanical strength and tensile shear resistance of the lap joints, as shown in Figure 7. After surface cleaning of stainless steel at 5000 mm/s and 1064 W laser power with optimal parameters, the tensile shear force of the weld after welding was increased by 54%.

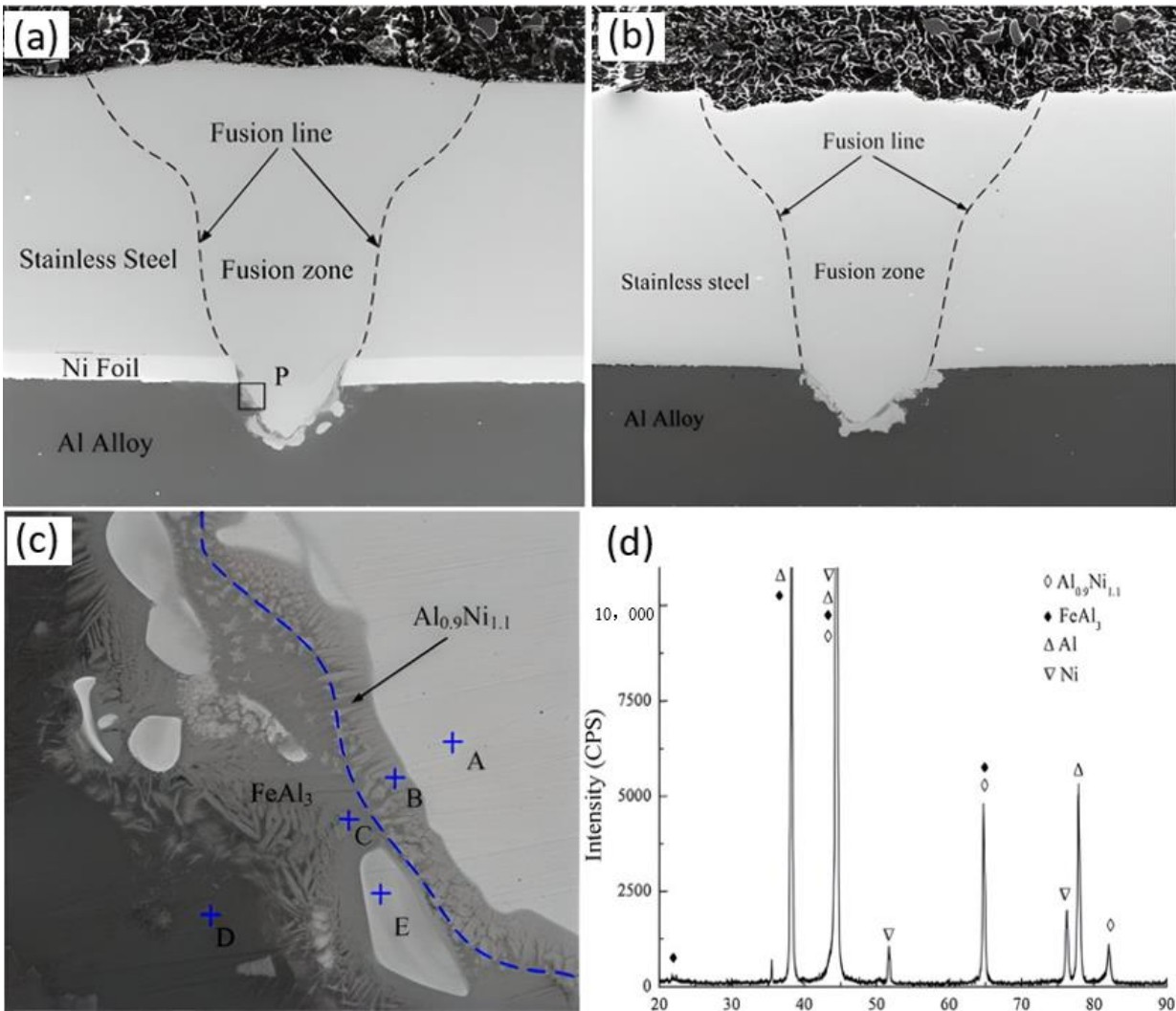

**Figure 6.** Microstructures of stainless steel/aluminum alloy joint. (**a**) Cross-section of the joint with Ni foil, (**b**) cross-section of the joint without Ni foil, (**c**) interfacial microstructures and (**d**) XRD patterns at the reaction zones. Reprinted with permission from Ref. [32]. Copyright 2012, Elsevier.

Chelladurai et al. [35] studied the welding of 3 mm 1050 aluminum and 0.25 mm steel sheet. The weld adopted the overlapping structure, the top was thin nickel steel, and the bottom was thick aluminum. They found that in laser welding, adding a wobble to the beam could effectively improve the shape of the welded joint. When the wobble amplitude was 0.2, 0.4 and 0.6 mm, there were visible cracks on the surface; when the wobble amplitude was 0.8–1.2 mm, there were slight spatters on the weld but almost no cracks on the weld surface, as shown in Figure 8; when the wobble amplitude was 0.2–0.4 mm, serious cracks appear, while the wobble amplitude of 0.6 and 0.8 mm shows slight cracks. When the wobble amplitude was 1 and 1.2 mm, there was no crack in the weld, as shown in Figure 9. A small wobble weld (<0.4 mm) showed a high degree of mixing of iron and aluminum, resulting in a large IMC phase forming in the top and middle regions. The FeAl phase mainly existed in the top, steel side and some rich Al $Fe_4Al_{13}$ on the weld/Al side. When the wobble amplitude was greater than 0.6 mm, the top of the weld is mainly Fe solid solution structure, and there was a small amount of Al, $Fe_4A_{13}$ and $Fe_2Al_5$ phase IMC at the bottom of the weld. When the wobble amplitude was greater than 0.8 mm, the melting area was larger and the formed weld had fewer IMC phases, so there was lower resistance. On the contrary, small wobble welds exhibited higher resistance because of more IMC phases, less contact area and more cracks.

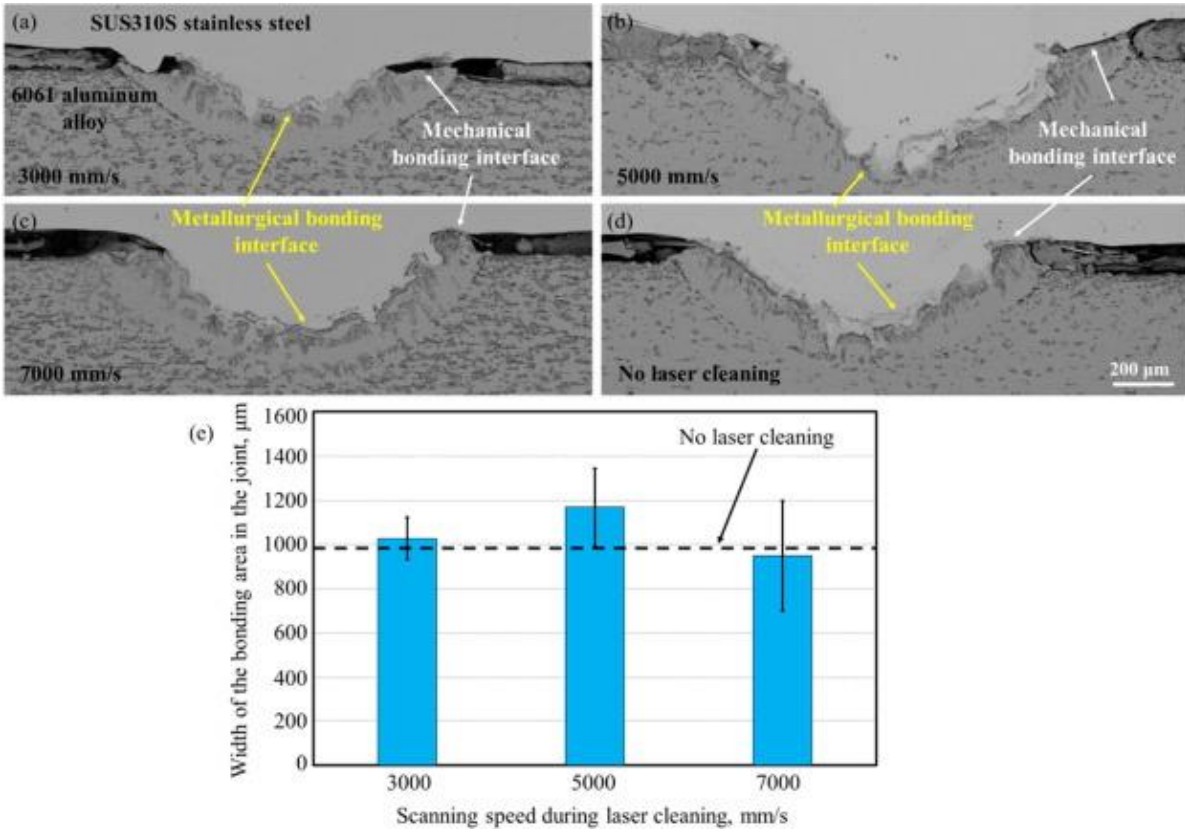

**Figure 7.** (**a–d**) Interface morphology at scanning speeds of 3000 mm/s, 5000 mm/s, 7000 mm/s and 0, respectively during laser cleaning. (**e**) Length of bonding area in the welded joint at different scanning speeds. Reprinted with permission from Ref. [34]. Copyright 2021, Elsevier.

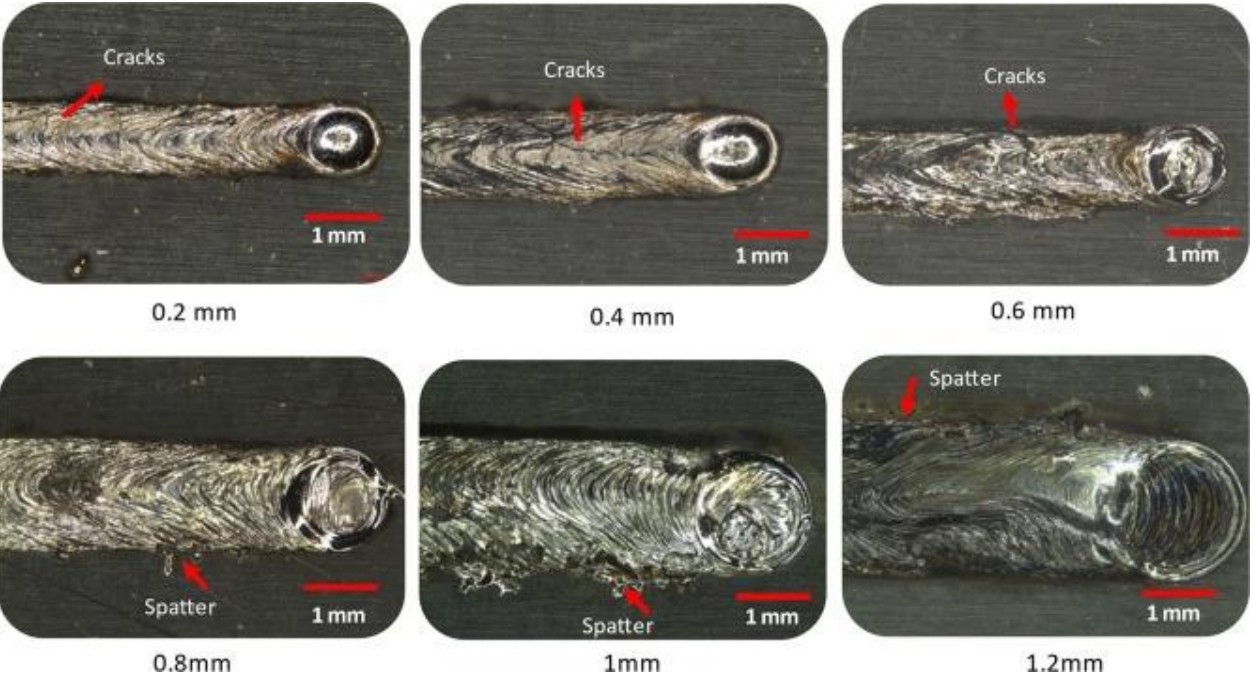

**Figure 8.** Surface images of welds of wobble amplitudes 0.2 to 1.2 mm [35].

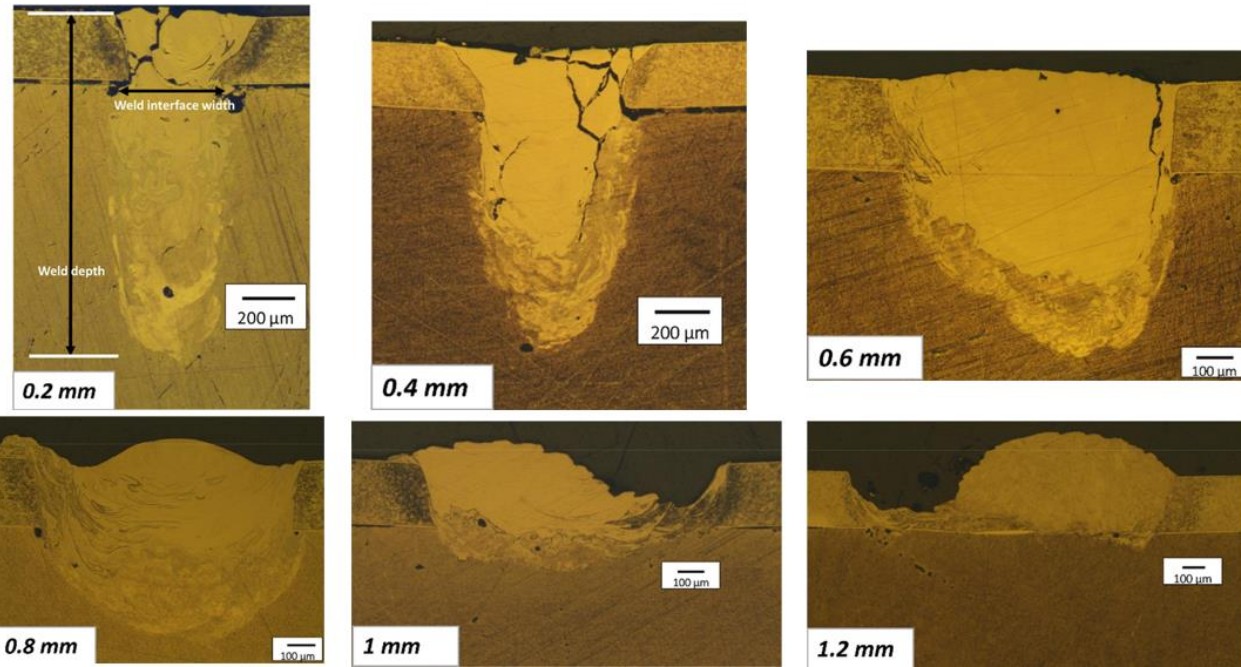

**Figure 9.** Weld macrostructures in different wobble amplitudes of 0.2 mm to 1.2 mm [35].

In order to improve the mechanical properties of aluminum and steel joints during welding, Sierra et al. [36] added Al-12Si for laser wire filling welding when welding AA6016 and DC04 steel, and they found that Si had a certain effect on the growth of Fe–Al intermetallic compounds. Fe–Al–Si intermetallic compounds with better mechanical properties were formed in the weld. Zhang et al. [37] also found that when laser welding H220YD steel and AA6016 aluminum used Al-5Si containing Si as interlayer, $Al_8Fe_2Si$, $\theta$-$Al_{13}Fe_4$ and $\xi$-$Al_2Fe$ intermetallic compounds of a certain thickness could be formed. When the thickness is greater than 10 μm, the joint strength decreases. Xia et al. [38] found that when welding 6061-T6 aluminum and DP590 steel, adding a sandwich containing Si could effectively reduce the laser power required for laser welding, and the formed intermetallic compounds containing Si had higher strength and shape, which could effectively improve the weld performance.

Reviewing the research in recent years, the laser welding of aluminum and steel has made great technical progress, but there is still a distance from the actual large-scale wide application, mainly because the mechanical properties of the joint are still insufficient. Progress has been made in process parameters and welding methods, but the formation of intermetallic compounds still needs to be solved, and there are other welding defects that need to be solved, such as pores and cracks. Table 2 summarizes the research on laser welding of aluminum and steel dissimilar metals.

## 2.2. Copper and Aluminum

Due to the differences in the melting point, thermal conductivity and thermal expansion of the two metals, the welding of copper–aluminum joints poses a major challenge [39–43]. Table 3 shows the main intermetallic compounds that can be formed between Cu and Al. Cu and Al in the welding process can form $Cu_2Al$, $Cu_4Al_3$, $CuAl$, $Cu_9Al_4$ and other intermetallic compounds [44]. The formation of these intermetallic compounds will greatly affect the microstructure and mechanical properties of the weld between Cu and Al [40,45–48]. Heideman et Al. [45] found that Cu and AL were welded with friction stir welding. With friction stir welding, which had the characteristics of low heat input, various intermetallic compounds would be produced in the welded joints. Abbasi et al. [47] found when welding Cu and Al using cold roll welding that although various intermetallic compounds such as $Cu_3Al$, $Cu_4Al_3$, $CuAl$ and $CuAl_2$ existed in the welded joints, the

growth rate of these intermetallic compounds was lower than that of friction stir welding. Laser welding has the characteristics of high energy density, fast welding speed and narrow heat-affected zone, which can further reduce the generation of intermetallic compounds. Moreover, when welding Cu and Al, filling silver, nickel, tin and other filler materials between the joints can also effectively reduce the formation of brittle phases [40–42].

**Table 2.** Summary of research conducted on laser beam welding of steel and aluminum.

| No. | Materials | Optimum Laser Parameters | Main Outcomes | Intermetallics | Ref. (year) |
|-----|-----------|--------------------------|---------------|----------------|-------------|
| 1 | 1060 Al 316L stainless steel | Power: 285 W Speed: 4 mm·s$^{-1}$ | The mechanical properties of the joint are related to the penetration depth | Not reported | [29] (2016) |
| 2 | 5754 Al 301 stainless steel | Power: 2 kW Speed: 1.4 m·min | Applying magnetic field can reduce grain size and stabilize weld quality | $Fe_2Al_5$ $FeAl_3$ | [30] (2016) |
| 3 | Low carbon steel st14 5754 Al | Power: 200 W Peak power: 1.43 kWSpeed: 5 mm/s | Increasing pulse time, pulse peak power and overlapping factor will result in the formation of more intermetallic compounds in the weld | $Fe_2Al_5$ $FeAl_3$ $FeAl_2$ | [31] (2010) |
| 4 | 201 stainless steel 5052 Al | Not reported | The addition of nickel interlayer helps to improve the metallurgical reaction of aluminum and iron, forming $Al_{0.9}Ni_{1.1}$ to improve the mechanical properties of the weld | $Fe_2Al_5$ $FeAl_3$ $Al_{0.9}Ni_{1.1}$ | [32] (2012) |
| 5 | Press-hardened steel 5052 Al | Power: 1.2 kW Speed: 12 mm/s | Using laser offset welding can improve the mechanical properties of weld | $Fe_2Al_5$ $Fe_4Al_{13}$ | [33] (2020) |
| 6 | 310S stainless steel 6061 Al | Power: 2.4 kW Speed: 1.5 m/min | Stainless steel surface helps to improve weld quality with laser cleaning before welding | Not reported | [34] (2021) |
| 7 | Hilumin steel 1050 Al | Power: 600 W Speed: 60 mm/s | With the increase in laser swing amplitude, the depth of weld decreases linearly, and the severity of weld cracking decreases significantly | $Fe_2Al_5$ $Fe_4Al_{13}$ $FeAl_2$ | [35] (2022) |
| 8 | DC04 steel 6016-T4 Al | Power: 2 kW Speed: 1 m/min | The strength of the assemblies is shown to increase linearly with the reaction layer width | Not reported | [36] (2008) |
| 9 | H220YD 6061 Al | Power: 2600 W Speed: 1 m/min | By laser filling wire welding, the fused aluminum alloy and the filling wire can be brazed to galvanized solid steel | $Al_8Fe_2Si$ $Al_{13}Fe_4$ $Al_2Fe$ | [37] (2013) |
| 10 | DP590 6061-T6 Al | Power: 2 kW Speed: 0.5 m/min | The joint produced with the AlSi5 filler metal had the highest tensile strength and largest fracture displacement. | $Fe_2Al_5$ $FeAl_3$ $Fe_2Al_8Si$ $Fe(Al,Si)_3$ | [38] (2018) |

**Table 3.** Properties of important intermetallics between Al and Cu.

| Phase | Cu Content (at.%) | Structure | Microhardness (HV) | Density (g/cm$^3$) | Specific Resistance (μΩ cm) |
|-------|-------------------|-----------|--------------------|--------------------|-----------------------------|
| $CuAl_2$ | 33 | Body-centered tetragonal | 630 | 4.34 | 8 |
| $CuAl$ | 51 | Body-centered orthorhombic | 905 | 5.13 | 11.4 |
| $Cu_4Al_3$ | 55.5 | Monoclinic | 930 | NA | 12.2 |
| $Cu_9Al_4$ | 66 | Body-centered cubic | 770 | 6.43 | 14.2 |

Ali et al. [49] used 1050Al 0.75 mm thick and coated 70 μm nickel (Ni) thin layer of 99.5 mm thick and 1.5 mm thick AA40 aluminum alloy copper for lap welding. They found that the weld width increased with the increase in laser power and decreased with the increase in welding speed. With the increase in heat input, the depth of the weld deepened continuously towards the same side, and there were defects such as cracks and pores in the weld. Under high temperature conditions, a large number of Al–Cu eutectic alloys ($\alpha$-Al + $Al_2Cu$) existed in the form of dendrites in the fusion zone of the weld, as shown in Figure 10. The high temperature also made the highly brittle $Al_4Cu_9$ phase

distributed in the weld, making the brittleness of the weld become higher. The heat input also significantly affected the contact resistance of the weld.

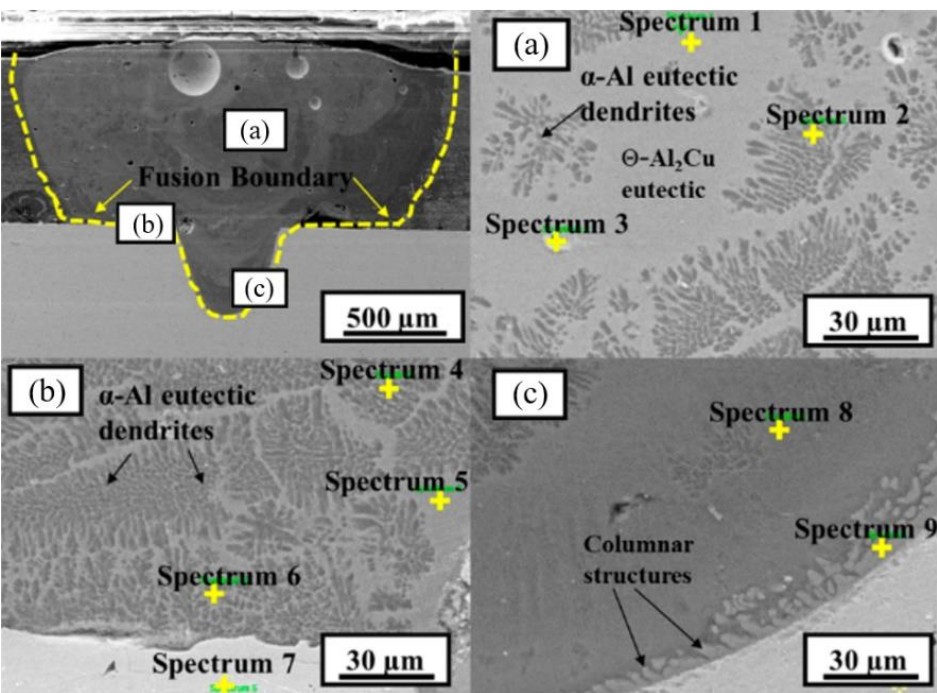

**Figure 10.** SEM images (power = 1500 W, welding speed = 30 mm/s); (**a**–**c**) are the magnified SEM images of microstructures at location **a**, **b** and **c** in weld fusion zone, respectively [49].

Lee et al. [50] used Al and Cu with purity of 99.99% as electrode samples. Al and Cu were used as the upper part of the lap joint for comparative test. A large amount of CuAl$_2$ was formed in the weld, and obvious $\alpha$(Cu) $\alpha$ phase, CuAl$_2$ phase and Al + CuAl$_2$ phase could be seen, as shown in Figure 11b,c. When welding with Al as the upper material, Cu was evenly distributed throughout the area except for the area where molten Cu penetrates the Al melt. When Cu was used as the upper material, the Cu mixing zone was distributed along the lower Al layer, because Cu mixing was heavier than molten Al, so it diffuses downward, as shown in Figure 11a. Further analyzing the effect of welding speed on welding quality, they found that CuAl$_2$, Cu$_9$Al$_4$ and CuAl intermetallic compounds could be observed in the weld at a welding speed of 10 m/min. At a higher welding speed of 50 m/min, the formation of intermetallic compounds was inhibited. In addition, with the increase in welding speed, the tensile strength was increased, when the welding speed was 50 m/min, the tensile strength of the top aluminum reached 160 MPa, and the tensile strength of the bottom aluminum reached 205 MPa.

Hailat et al. [51] studied the continuous laser welding of 3003 aluminum and 110 copper. They welded two sets, one with tin as a sandwich and the other without tin. In welds with Sn interlayers, large pores could be seen in aluminum. However, the breaks occur far away from these pores, so they did not appear to affect joint strength, and the welds of tin-filled metals exhibit better bond shear strength, possibly due to the formation of Cu$_6$Sn$_5$ and Cu$_3$Sn.

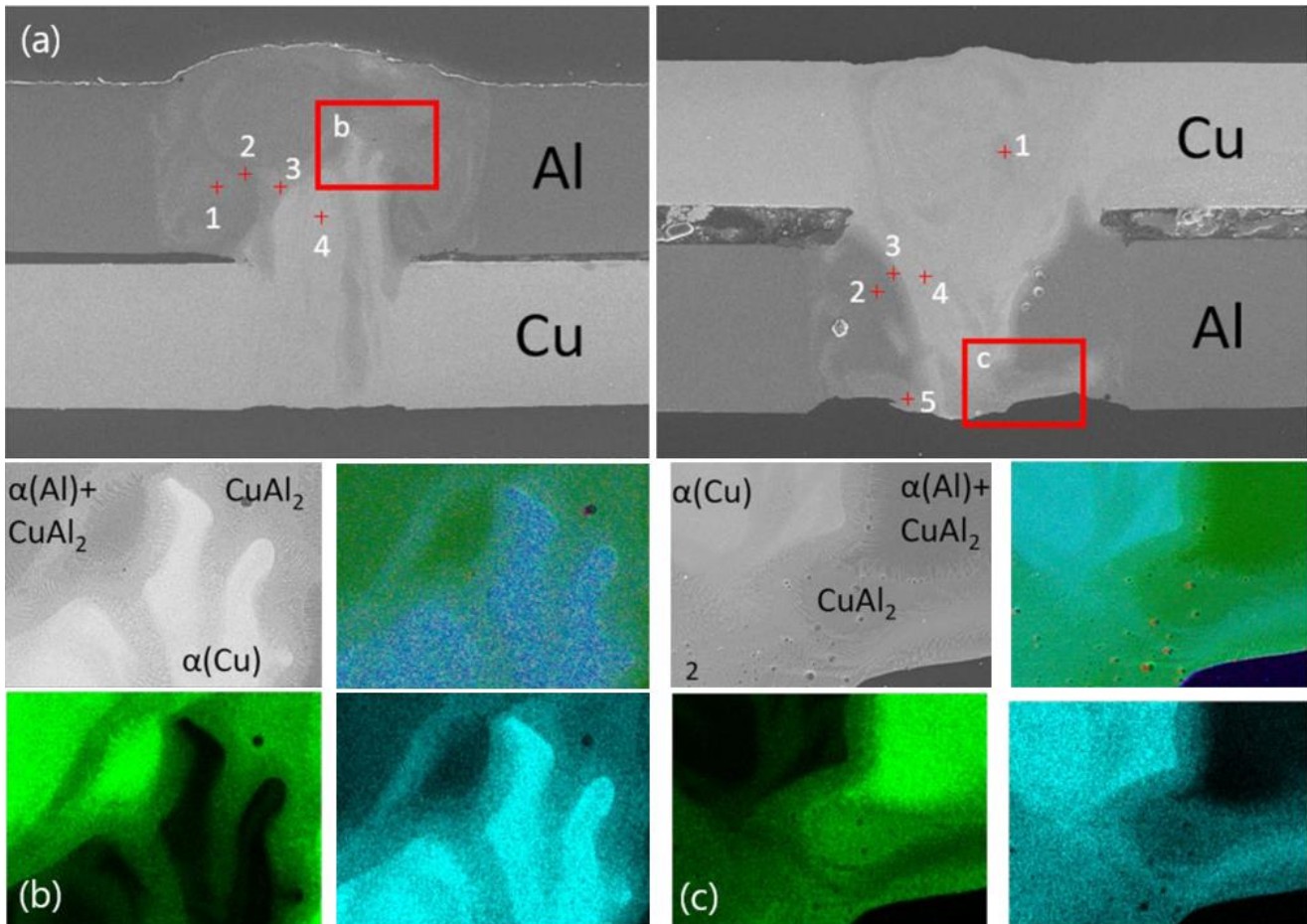

**Figure 11.** (**a**) Scanning electron microscopy images of the welded specimens. (**b**,**c**) EDS mapping images showing the distribution of Al and Cu to observe phase formation at a welding speed of 400 mm/s with: (**b**) Al as the upper layer and (**c**) Cu as the upper layer [50].

Xue et al. [52] observed the microstructure of the intermediate layer in laser welded copper aluminum lap joints. Tensile shear tests were conducted, and the fracture morphology was analyzed using SEM and EDS. The results indicate that there are several different regions in the interlayer of the weld seam, with different morphological and compositional characteristics, as shown in Figure 12. Banded and cellular structures were observed in the hypereutectic zone. The eutectic zone had a layered structure, the narrowest and thinnest. The thickest and widest dendritic structure was obtained in the hypoeutectic region. The joint fracture developed in the dendritic hypoeutectic zone, and the fracture mode was a combination of brittleness and shear. The maximum shear load of Cu–Al joints decreases with the increase in primary dendrite arm spacing and the growth of secondary dendrites in the hypoeutectic region, which was caused by an increase in laser power. Fusion welding of aluminum and copper, being dissimilar materials, is difficult, because brittle intermetallic compounds are formed in the welding zone, the weldability is poor, and the chemical, mechanical and thermal properties of welded joints are different. Due to the inevitable formation of brittle intermetallic compounds, the connection of aluminum and copper plates presents a metallurgical challenge. Therefore, it is necessary to effectively inhibit the formation and growth of Al–Cu intermetallic compounds. For the welding of dissimilar aluminum and copper sheets, there is no systematic work to reduce these defects.

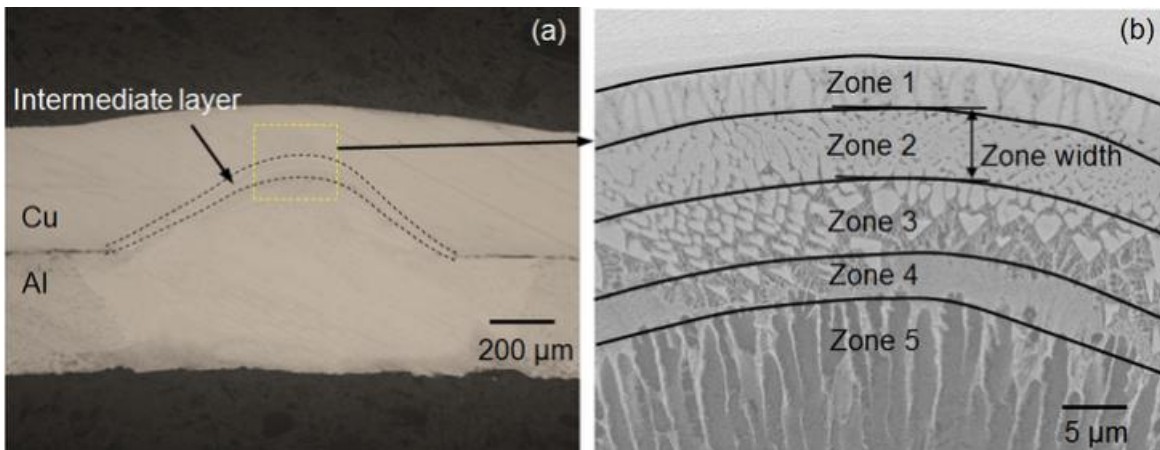

**Figure 12.** Micrograph of Cu–Al joint cross-section. (**a**) Optical micrograph; (**b**) SEM image of intermediate layer marked by the dashed square box in (**a**). Reprinted with permission from Ref. [52]. Copyright 2013, Elsevier.

Lee et al. [40] focused on the effect of welding speed on the quality of single-mode fiber laser lap welded joints of aluminum and copper sheets. They found that the intermetallic compounds are easy to form in a wide area at the welding speed of 1 kW and 10 m min$^{-1}$, while the intermetallic compounds were greatly reduced at the higher welding speed of 50 m min$^{-1}$. The width of the intermetallic compound decreased sharply to about 5 mm. According to the results of tensile shear test, the different loads and advantages of Al (upper) copper (lower) and copper (upper) process (lower) welding joints were almost equal to the welding speed of 50 m min$^{-1}$ in most cases, and the loads of different welded joints were higher than those of similar devices, which were almost equivalent to the similar speed of Cu–Cu, and the tensile shear strength of the weld metal increases with the increase in the welding speed. Therefore, by inhibiting the formation of intermetallic compounds, solid laser welded joints could be produced at extremely high welding speed.

When welding Cu and AA4047 aluminum, Mai et al. [48] found that a weld without cracks could be obtained by shifting the laser to the aluminum side by 0.2 mm, and the weld had a higher hardness than the base material. Weigl et al. [53] used AlSi$_{12}$ as filler material for laser welding of pure copper and pure aluminum, respectively. They found that both AlSi$_{12}$ and CuSi$_3$ intermetallic compounds in the weld increased the ductility of the joint, and AlSi$_{12}$ with a higher Si content was more effective.

At present, the main research direction of welding between aluminum and copper has been the optimization of process parameters and the use of interlayer, and the subsequent research can add beam oscillation. Although the optimization of process parameters has yielded preliminary results, the potential formation of intermetallic compounds still needs further research. Table 4 summarizes the current research on aluminum and copper dissimilar laser welding.

### 2.3. Copper and Steel

In the welding of electric vehicle batteries, there are many types of welding between copper and steel, and Table 5 shows the room temperature properties of copper and iron. From the table, it can be seen that there are significant differences in the physical properties of copper and iron, especially the differences in melting temperature and thermal conductivity, making welding the two metals challenging [54]. In the Fe and Cu phase diagrams, there is a wide metastable miscibility gap at high temperatures [55]. In laser welding of steel and copper, liquid phase separation is a common feature due to the separation of undercooled Fe–Cu liquid into droplets of iron and copper [56]. Another major problem is that hot cracks appear in the welding zone or the heat-affected zone (HAZ) of the steel due to the penetration of Cu into the grain boundary [57].

**Table 4.** Summary of research conducted on laser beam welding of steel and aluminum.

| No. | Materials | Optimum Laser Parameters | Main Outcomes | Intermetallics | Ref. (Year) |
|---|---|---|---|---|---|
| 1 | Cu99.5% AA 1050 | Power: 1600 W Speed: 30 mm/s | The greater the heat input, the more intermetallic compounds are generated. The resistance decreases as the welding speed decreases. | $Al_4Cu_9$ $Al_2Cu$ | [49] (2022) |
| 2 | Cu99.9% Al99.9% | Power: 2000 W Speed: 400 mm/s | The fracture after welding is mainly on the copper side. | $CuAl_2$ | [50] (2022) |
| 3 | Cu 110-H00 Al 3003-H14 | Power: 500 W Speed: 1 m/min | Adding tin alloy foil as interlayer can improve the mechanical properties of weld. | Not reported | [51] (2011) |
| 4 | T2 Cu 1060 Al | Power: 1450 W Speed: 100 mm/s | The microstructure of the subeutectic zone will greatly affect the shear resistance of the joint. | CuAl $CuAl_2$ | [52] (2014) |
| 5 | Cu99.57% A1050 Al | Power: 1 Kw Speed: 10 m/min | Increasing the welding speed helps to reduce the content of intermetallic compounds in the weld. | CuAl $CuAl_2$ | [40] (2013) |
| 6 | Oxygen-free Cu 4047 Al | Not reported | Controlling the melting ratio of metals is an important factor for defect-free welding of dissimilar metals. | Not reported | [48] (2004) |
| 7 | Pure Cu Pure Al | Not reported | The aluminum filler alloy AlSi12 produces a more uniform elemental mixture and a significantly enhanced ductility. | Not reported | [53] (2011) |

**Table 5.** Summary of the room temperature properties of Al, Cu, Fe and Ni.

| Metal | Melting Temperature (K) | Boiling Temperature (K) | Density (Kg m$^{-3}$) | Thermal Conductivity (W m$^{-1}$ K$^{-1}$) | Thermal Expansion Coefficient ($10^6$K$^{-1}$) |
|---|---|---|---|---|---|
| Fe | 1809 | 3133 | 7870 | 78 | 12.1 |
| Al | 933 | 2739 | 2700 | 238 | 23.5 |
| Cu | 1356 | 2833 | 8930 | 398 | 17 |
| Ni | 1728 | 3188 | 8900 | 89 | 13.3 |

Joshi et al. [58] proposed the idea of using laser offset to weld copper and stainless steel butt joints. They shifted the laser to the stainless steel side, and there were obvious stainless steel particles in the weld after welding, and there were obvious solidification cracks after welding. Meng et al. [59], when using laser-arc composite welding to weld T2 copper/304 stainless steel, found that the welding position tilted to the pure copper side can effectively improve the welding quality. They found that when the welding position shifted to the pure copper side, the melting of the stainless steel gradually decreased, and the weld after welding had obvious depression on the surface and became full on the surface, as shown in Figure 13. They finally measured the tensile strength of the weld to 215 MPa, which was close to the tensile strength of T2 copper. Chen et al. [60], using laser welding to weld copper/stainless steel joints, also found that the laser offset to the stainless steel side could create a brazing effect, and they believed that the copper content in the joint should be limited to reduce the generation of cracks.

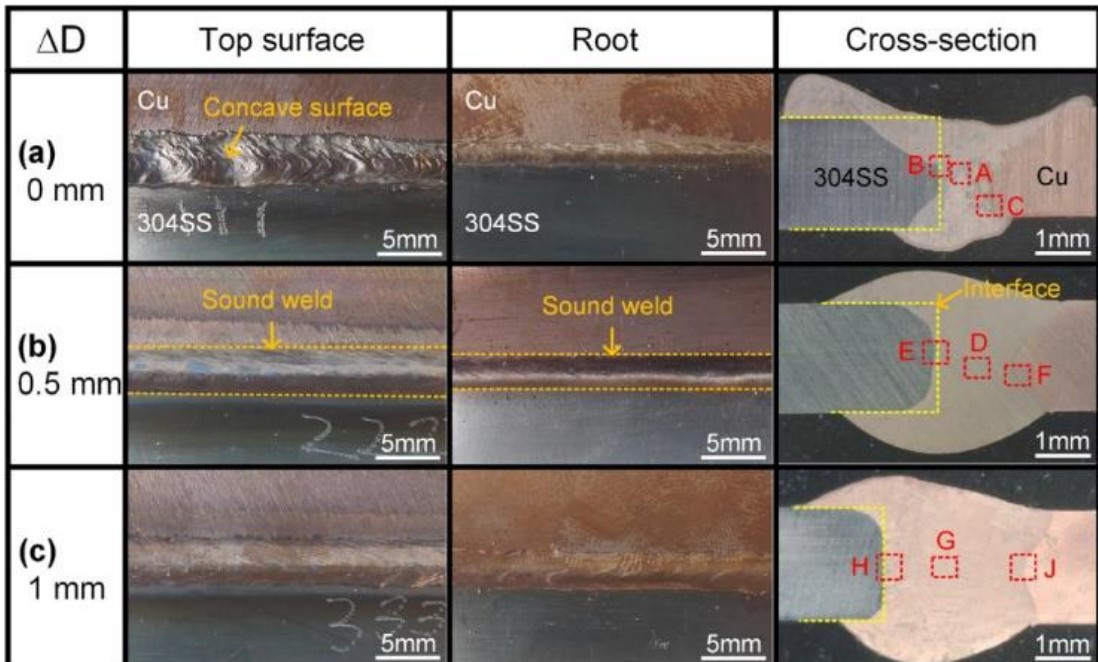

**Figure 13.** Surface and cross-sectional morphologies of the Cu/304SS dissimilar joints: ΔD = 0 mm, ΔD = 0.5 mm and ΔD = 1 mm. (**a**) Laser power = 3 kW, Arc current = 60 A, Welding speed = 1 m/min, Laser offset = 0 mm, (**b**) Laser power = 3 kW, Arc current = 60 A, Welding speed = 1 m/min, Laser offset = 0.5 mm, (**c**) Laser power = kW, Arc current = 60 A, Welding speed = 1 m/min, Laser offset = 1.0 mm. Reprinted with permission from Ref. [59]. Copyright 2019, Elsevier.

Fei et al. [61] conducted laser welding experiments on pure copper and stainless steel under the condition of annular beam oscillation. It was found that the crack resistance of weld could be improved effectively by adding oscillation. As could be seen from Figure 14, after adding oscillation, the microstructure in Figure 14d was significantly finer than that in Figure 14a, and the grain microstructure at the grain boundaries also presented discrete distribution, and the grain direction changed from single direction to multi-direction. When ring oscillating laser welding was used, the structure changed from blocky structure to spherical structure compared with Figure 14e, and the (α + ε) structure was also evenly distributed in the welding area. The addition of beam shaking promotes solute migration and enhanced the consistency of plastic deformation in grains. Moreover, due to the larger range of action in the solidification process, the energy input and temperature field distribution were changed, and the thermal stress in the weld was also reduced. Under the action of beam vibration oscillation, Fe diffused into the Cu matrix, forming blocky (α + ε) and spherical (α + μ) structures, as shown in Figure 14f. In addition, due to the driving action, the dissolution amount of copper in the weld increased, and the microstructure gradient of the weld decreased.

The microstructure with the addition of beam oscillation, as shown in Figure 15, was significantly refined compared with that without beam oscillation, as shown in Figure 15a. During the solidification process, the increase in the number of grain boundaries significantly reduced the stress concentration at the grain boundaries and improved the deformation. After adding the beam vibration, because the microstructure was more refined, the composition distribution was more uniform, and the strength and shape of the joint were effectively improved. The tensile strength and elongation of the joint were 282.82 MPa and 2.68 mm, respectively. Compared with the joint without beam vibration, the strength was increased by 10.41% and the tensile lift rate was increased by 53.14%, as shown in Figure 16.

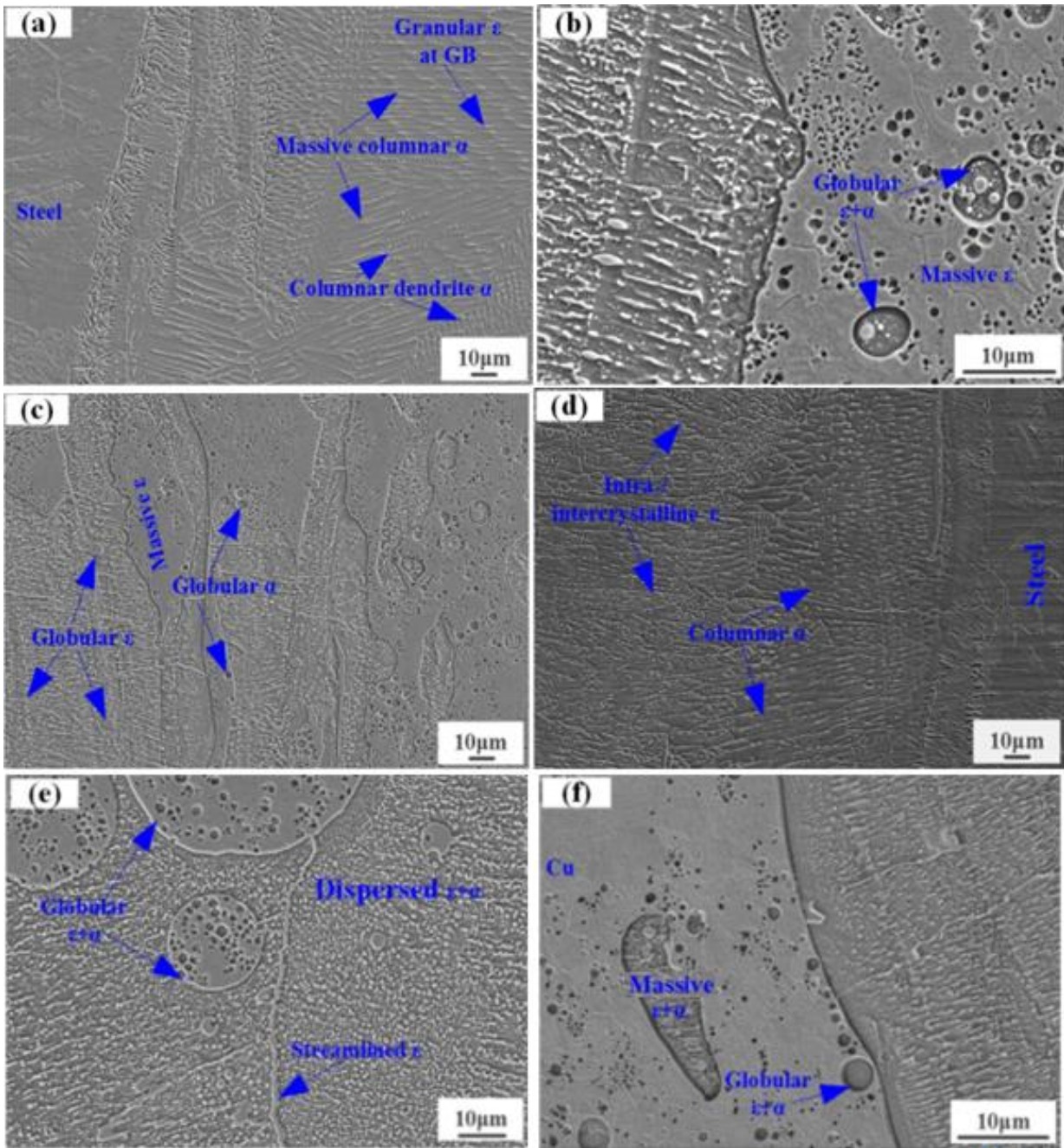

**Figure 14.** Microstructure of joints welded in the shortage of beam oscillation (**a–c**) and with beam oscillation (0.5 mm, 250 Hz) applied (**d–f**): (**a/d**) joint microstructures near the steel matrix; (**b/e**) joint microstructures in the middle weld; and (**c/f**) joint microstructures near the Cu matrix. Reprinted with permission from Ref. [61]. Copyright 2022, Elsevier.

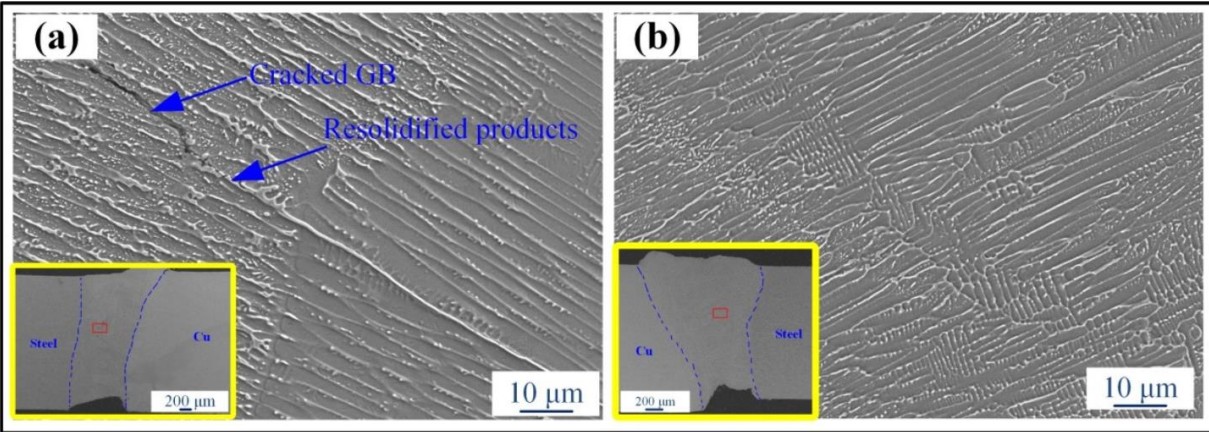

**Figure 15.** SEM micrographs of weld microstructures: (**a**) weld microstructures without exposure to beam oscillation and (**b**) weld microstructures at 0.5 mm, 250 Hz. Reprinted with permission from Ref. [61]. Copyright 2022, Elsevier.

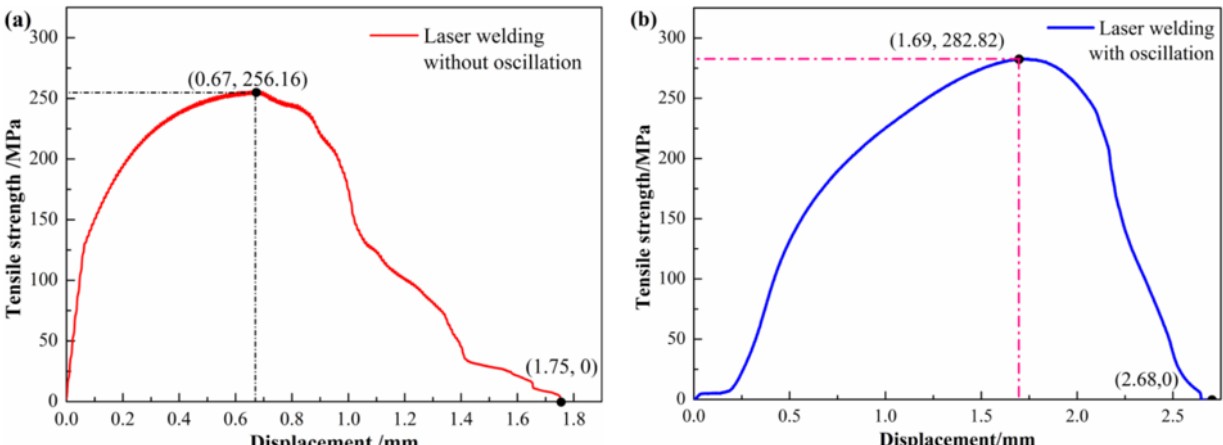

**Figure 16.** Tensile strength of joints welded: (**a**) with beam oscillation and (**b**) without beam oscillation. Reprinted with permission from Ref. [61]. Copyright 2022, Elsevier.

Mannucci et al. [62] welded 316L austenitic stainless steel with copper and found that the tensile property of the weld was mainly determined by the formation of heat-affected zone in solid copper. The melting zone of copper and stainless steel had a strongly asymmetrical shape, as shown in Figure 17, and the stainless part of the weld was much more developed and had an hourglass shape due to two eddies created by Marangoni convection. No heat-affected zone (HAZ) was formed in 316L side. The copper side of the weld was almost straight and bordered by a wide HAZ. Under the laser power, the Cu content in the melting zone increases with an increase in the laser power but never reached 50%, because Cu has a very high thermal diffusivity compared with 316L. Only the welded joint produced by the laser offset to the copper side had good performance.

Li et al. [63] found that the defects of laser welding between stainless steel and copper were mainly due to liquefaction cracking in the heat-affected zone of stainless steel and porosity in the fusion zone. Due to the presence of Fe–Cu compounds at the grain boundaries of the heat-affected zone, the bonding force between grains was affected and the sensitivity of weld cracks was increased. They found that the formation of liquefaction cracks could be divided into three stages, as shown in Figure 18. In the first stage, copper atoms permeated continuously along the grain boundaries, and cracks began to spread at the grain boundaries. In the second stage, Fe–Cu compounds accumulated at the grain boundaries, which led to the destruction of the bonding strength between grains and the initiation of cracks. In the third stage, with the increase in heat input in the laser welding

process, the thermal stress increased significantly, causing the small crack at the grain boundary to expand into a large crack. The length of the crack increased with the increase in the heat input, as shown in Figure 19. However, when the heat input reached 125 KJ/m, the crack length began to decrease. This was because the molten copper present in the crack has self-healing properties due to a further increase in temperature. Although this self-healing property exists, the heat input should also be reduced during welding to better control the welding quality. In addition, they found that the porosity was largely due to the instability of the keyhole during the welding process, independent of the liquefaction cracking of the heat-affected zone. The laser focus during welding could be shifted to the stainless steel side, which could change the flow of the liquid metal and increase the stirring effect, which could help eliminate pores.

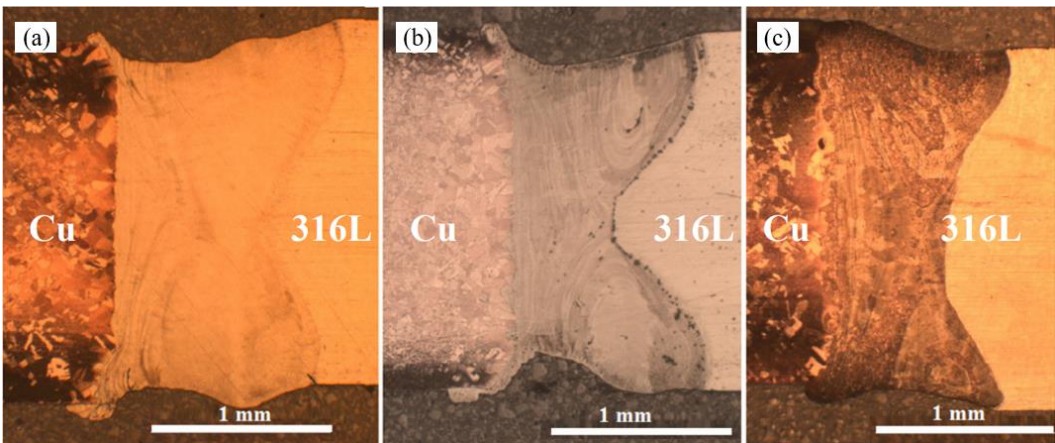

**Figure 17.** Typical weld cross-sections with beam shift: (**a**) 0.4 mm on steel (**b**) zero and 0.2 mm on copper and (**c**) 0.2 mm on copper [62].

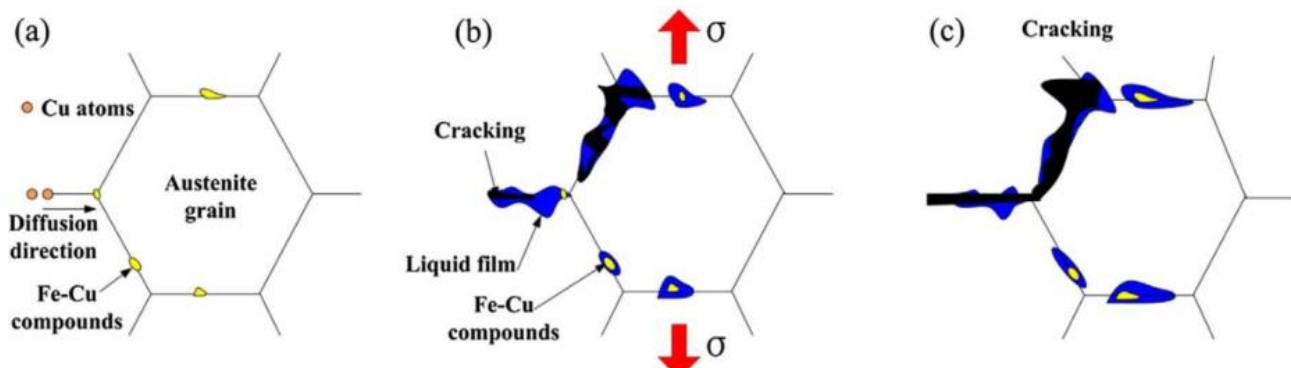

**Figure 18.** Liquation cracking model: (**a**) incubation, (**b**) initiation and (**c**) growth. Reprinted with permission from Ref. [63]. Copyright 2020, Elsevier.

Shen et al. [64] found that during laser welding of 316 stainless steel and oxygen-free copper, after the laser focus was shifted 0.4 mm to the copper side, no solidification cracks were detected in the weld after welding, and the copper content in the weld was 80%. Shaikh et al. [15] found in laser welding of copper and steel containing nickel coatings that the laser power, pulse opening time and frequency were positively correlated with the shear strength of the joint. And as the mixing degree of steel and aluminum increased, the strength of the joint increased.

Welding of butt joints of copper and steel has been studied. However, further studies of the microstructure of the weld zone and the interaction between the two materials are needed. Lap joints suitable for battery welding need to be paid more attention, and a

sandwich can be added when welding. Table 6 summarizes the studies conducted on different laser welding of steel and copper.

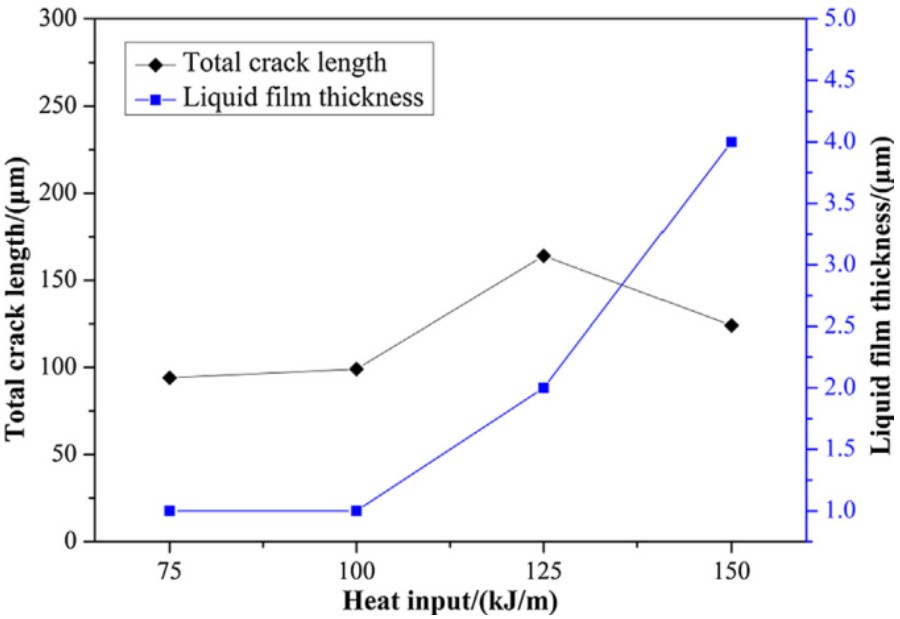

**Figure 19.** The correlation between the total crack length and heat input. Reprinted with permission from Ref. [63]. Copyright 2020, Elsevier.

**Table 6.** Summary of research conducted on laser beam welding of steel and copper.

| No. | Materials | Optimum Laser Parameters | Main Outcomes | Intermetallics | Ref. (Year) |
|---|---|---|---|---|---|
| 1 | 304L stainless steel ETP Cu | Power: 1000 W Speed: 300 mm/min | A defect-free bimetallic joint between Cu and SS can be obtained by laser beam welding technique with fusion welding mode. | Not reported | [58] (2019) |
| 2 | 201 stainless steel T2 Cu | Not reported | Welding of copper/steel when welding—brazing and fusion welding depends on the welding parameters. | Not reported | [60] (2013) |
| 3 | 304 stainless steel T2 Cu | Power: 4 kW Speed: 3 m/min | Adding beam oscillation in laser welding can refine grain and enhance weld mechanical properties. | Not reported | [61] (2022) |
| 4 | 316L stainless steel CW Cu | Not reported | Excessive laser power can lead to the formation of thermal cracks. | Not reported | [62] (2018 |
| 5 | 304 stainless steel T2 Cu | Not reported | The susceptibility to HAZ liquation cracking can be effectively lowered by controlling the heat input during laser welding. | $Cu_{40}Fe_{60}$ $Cu_xFe_{1-x}$ | [63] (2020) |
| 6 | 316 stainless steel oxygen-free Cu | Not reported | By adjusting welding parameters, weld defects such as shrinkage, porosity, solidification cracks, etc., can be eliminated. | Not reported | [64] (2004) |
| 7 | Steel-Hilumin Cu | Power: 60 W Speed: 500 mm/min | Mechanical strength of the joint is highly correlated with electrical resistance and corresponding temperature rise at the joint. | Not reported | [15] (2019) |

## 3. Welding of Battery Housing

The sealing of the battery housing, especially for hard cases, requires a high-quality weld, and since the electrical components in the case are very sensitive to heat input, this requires a low heat input [65]. Shown in Figure 20 is a Tesla 4680 battery. The top and bottom of the battery case need to be welded. Welding is required to have a stable weld depth,

and there can be no molten drop in the tank because the molten drop will destroy the very sensitive electrolyte. After welding, in addition to assuring that the weld has good mechanical properties, it is crucial to ensure that the battery shell cannot be deformed. Therefore, laser welding has the characteristics of fast speed, small heat input and small range of action, which is an ideal processing mode for this production step in battery production. The battery housing is mainly made of aluminum alloy and steel, so it is necessary to study the same metal welding of welding aluminum alloy and stainless steel separately.

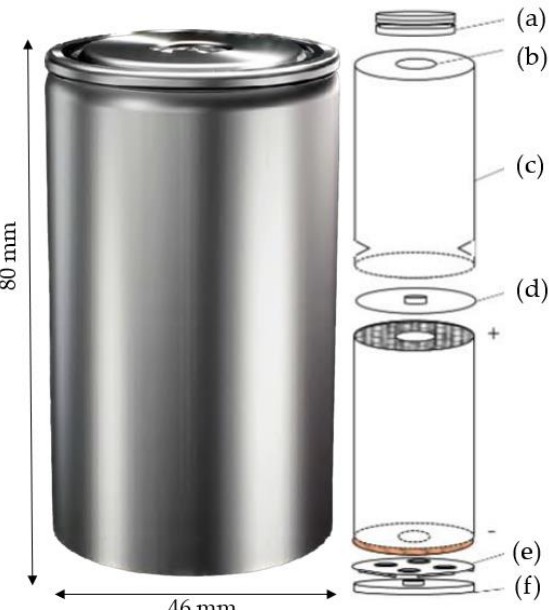

**Figure 20.** Tesla 4680 battery: (**a**) Pole cylinder, (**b**) front opening, (**c**) can, (**d**) positive collector plate, (**e**) negative collector plate and (**f**) cover plate.

### 3.1. Welding of Aluminum Battery Housing

Aluminum alloy has been widely used in various fields because of its advantages of light weight, high strength, good corrosion resistance and good formability such as in automobile manufacturing, aerospace, ship manufacturing, track manufacturing and so on. However, the welding of aluminum alloy has been facing various challenges because aluminum alloy has good thermal conductivity and high coefficient of thermal expansion, making it difficult to weld. Aluminum alloy welding is usually accompanied by cracks and pores and other welding defects [66–69]. The oxide film ($Al_2O_3$) and other organic impurities on the surface of aluminum alloy are easy to decompose at high temperature, which improves the porosity sensitivity of the weld. Gas phase components such as hydrogen are more soluble in the melt pool at high temperatures, and the faster cooling rate makes it difficult for them to escape and form pores [44].

In order to solve the problems of forming porosity and poor welding stability in the welding process of aluminum alloy, Wu et al. [70] proposed a new welding method. They used fiber laser with focused rotation and vertical oscillation to weld 1060 aluminum alloy. This method combined a vertical oscillation rotation of the focus along the direction of the beam. They found that when only vertical oscillation laser focusing welding was added, the width of the weld was relatively small, and with the increase in the oscillation amplitude, the shape of the weld surface first continuously improved, and when the oscillation amplitude continued to increase, the surface morphology of the weld began to gradually deteriorate. When the vertical oscillation amplitude was 2 mm, the weld surface formation was best, as shown in Figure 21. When the rotation radius was within the range of 0~0.45 mm, the formation of the weld surface gradually improved with the increase in the oscillation amplitude, and the surface formed had fish-scale ripples, which were dense, uniform and smooth. This indicated that focal rotation could improve the quality of weld

surface formation. Further observation of the depth of the weld showed that, as shown in Figure 22, when the rotation radius was less than 0.15 mm, it had little influence on the penetration depth of the welds using vertical oscillation laser welding and has little change compared with no addition. The morphology of the weld section was thin and long nail shape. When the rotation radius of the laser was 0.45 mm, the weld section was shallow but wide nail shape, and the width of the weld and the middle part are relatively large. The weld morphology of focus rotating laser and vertical oscillation laser was similar to that of focus rotating laser welding only. It could be said that the depth of the weld was mainly affected by the amplitude of the vertical oscillation laser, and the weld appearance was mainly affected by the rotation radius of the laser focus rotation. They also found that adding the rotation of the focus during the welding process could effectively reduce the formation of porosity, and the larger the rotation radius, the lower was the porosity. When the rotation radius was 0~0.45 mm, the porosity decreased with the increase in the rotation radius. When the rotation radius was 0.45 mm, the porosity of the weld was reduced by 91% compared with the non-focused rotation and non-vertical oscillation, as shown in Figure 23.

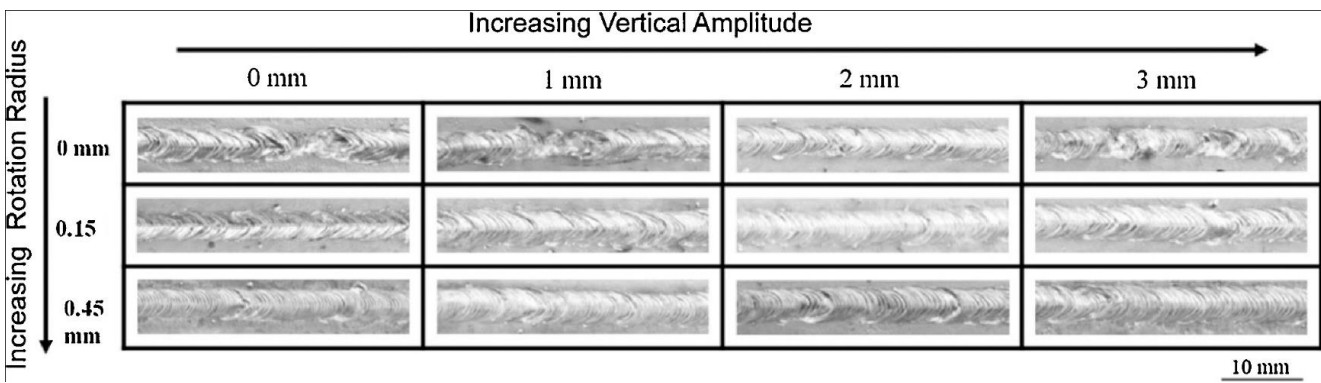

**Figure 21.** Surface morphologies of laser beam focus rotation and vertical oscillation welding ($f_{rot} = f_{osc} = 100$ Hz). Reprinted with permission from Ref. [70]. Copyright 2018, Elsevier.

Mauritz et al. [71] also adjusted the laser. On the basis of adding beam vibration, they applied the laser spot beam shaping technology to divide individual laser spots into four single beams with equal power, and then interacted with the material in a processing area, as shown in Figure 24. The dynamic situation of the molten pool was observed by high-speed photography, as shown in Figure 25. They found that the keyhole size of the molten pool significantly increased when the multi-focus technology was adopted. Compared with the average area of 0.13 mm$^2$ during single-beam welding, the average area of the multi-focus technology increased by 10.6 times, reaching 1.51 mm$^2$. Further dynamic observation showed that the stability of metal vapor in the molten pool is increased with the increase in keyhole area. Compared with single point welding, the keyhole of multi-focus method is constantly opened. Moreover, the standard deviation of keyhole size is effectively reduced by the multi-focal technique, and the area fluctuation of keyhole is reduced by 7.1% from 54.3% in single point welding. Compared with the single-point process, the multi-focus method doubles the length of the weld pool, which also allows the energy input to be more evenly distributed in space, effectively reducing the splash during welding. X-ray observation was performed on the section after welding, as shown in Figure 26. Compared with single-point welding, the size of pores in the weld was significantly reduced when the multi-focus technology was adopted for welding, so the new technology greatly improved the quality of the weld.

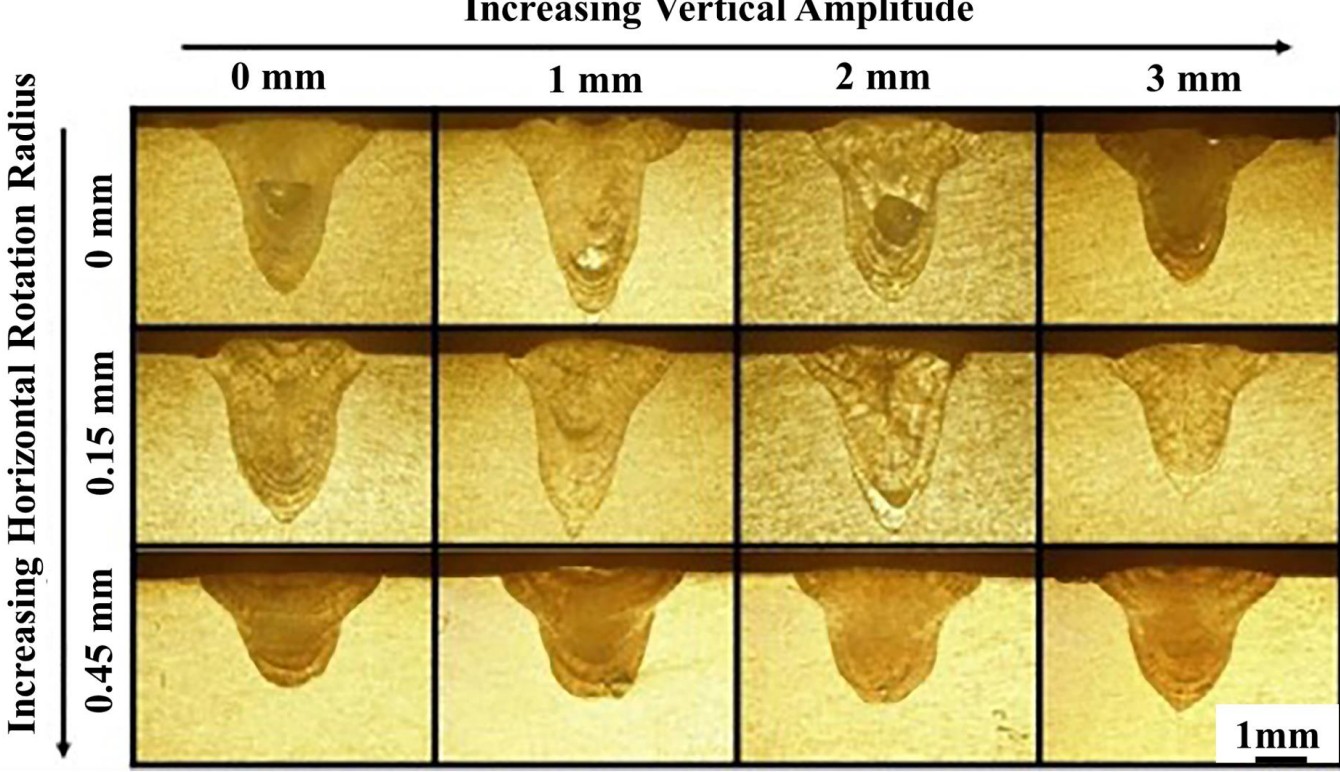

**Figure 22.** Effect of rotation radius and vertical oscillation amplitude on weld cross-section morphology ($f_{\text{rot}} = f_{\text{osc}} = 100\,\text{Hz}$). Reprinted with permission from Ref. [70]. Copyright 2018, Elsevier.

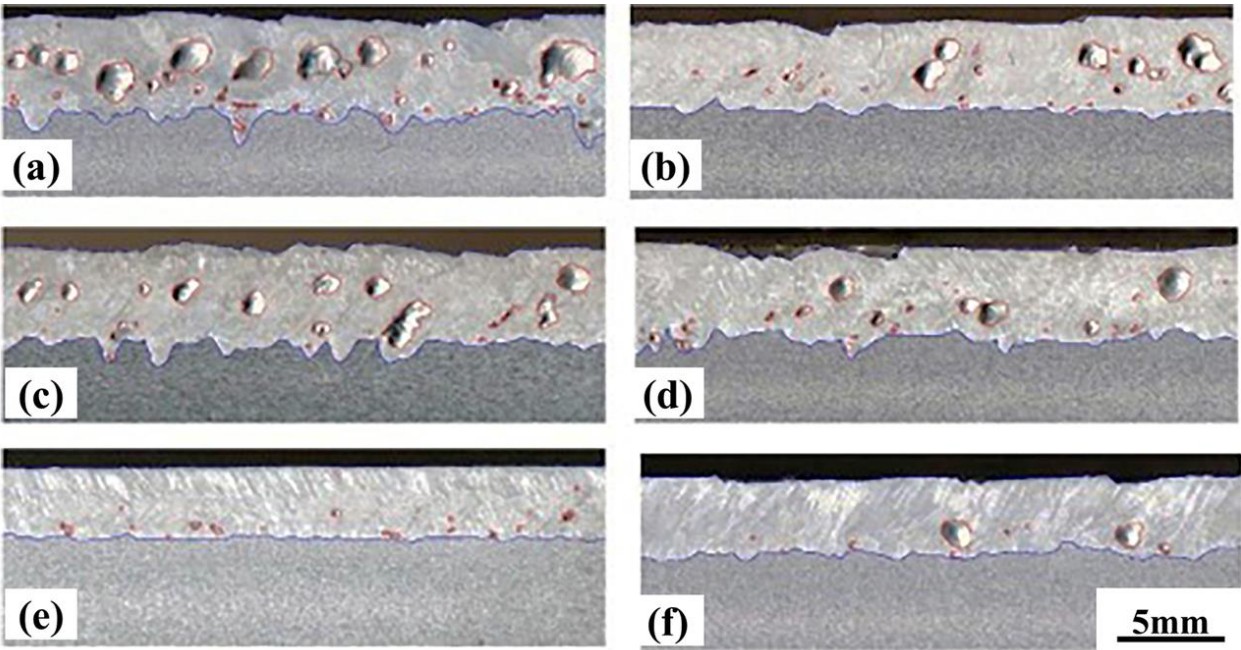

**Figure 23.** Pores in longitudinal section of welds under different rotation radius and vertical amplitude: (**a**) No oscillation, (**b**) $A_{osc} = 2\,\text{mm}$, (**c**) $R_{rot} = 0.15\,\text{mm}$, (**d**) $R_{rot} = 0.15\,\text{mm}$ and $A_{osc} = 2\,\text{mm}$, (**e**) $R_{rot} = 0.45\,\text{mm}$ and (**f**) $R_{rot} = 0.45\,\text{mm}$ and $A_{osc} = 2\,\text{mm}$. Reprinted with permission from Ref. [70]. Copyright 2018, Elsevier.

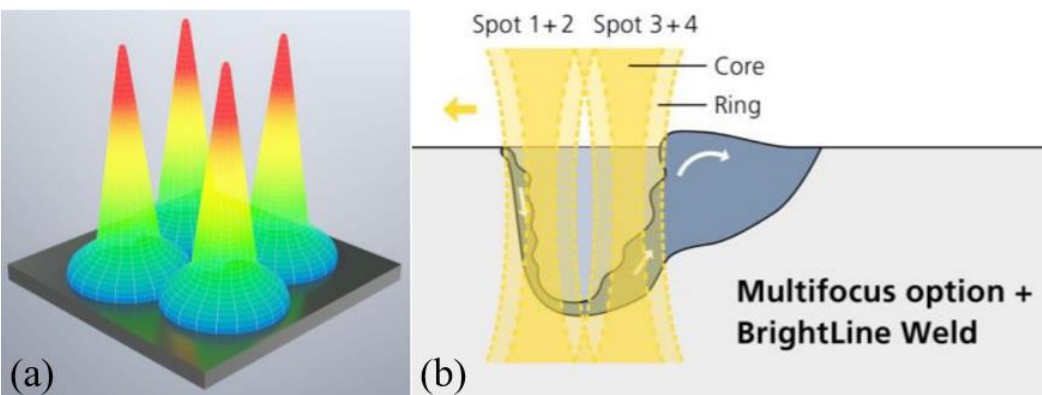

**Figure 24.** (**a**) Schematic diagram of laser power distribution of four spots formed by multi-focusing. (**b**) Schematic diagram of laser spot acting on molten pool [71].

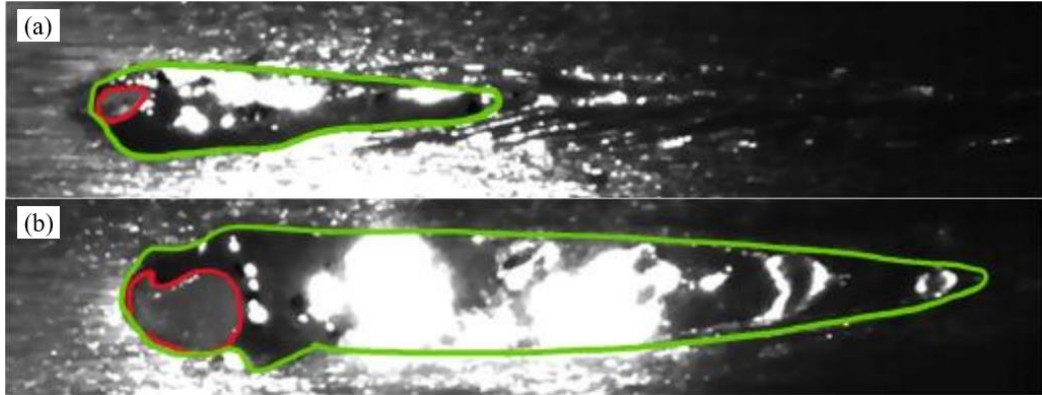

**Figure 25.** (**a**) Single spot welding with beam shaping. (**b**) Multi-focus approach with stabilized keyhole. [red: current keyhole size; green: current weld pool size] [71].

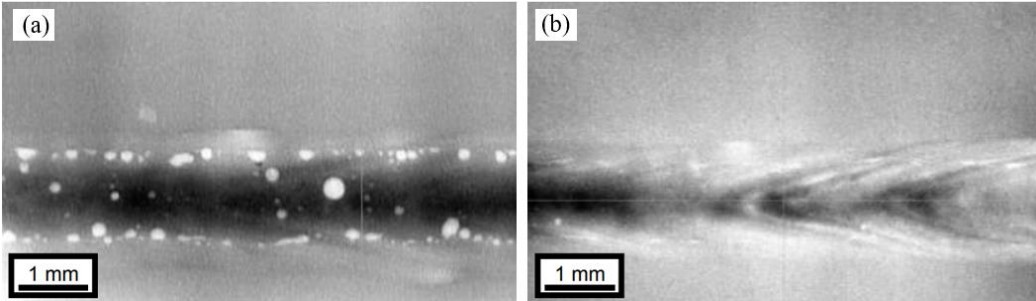

**Figure 26.** (**a**) X-ray image of single spot welding with beam shaping shows significant porosity in the weld seam. (**b**) X-ray image of multi-focus welding shows substantial reduction of the porosity in the weld seam [71].

Prieto et al. [72] shaped the laser spot and formed the laser into six shapes as shown in Figure 27 for welding experiments on AA3003. By comparison, they found that when the laser power was between 2.4~3.6 kW, the welding speed was 18 m/min, and the shape frequency was up to 1 MHz, and the best weld formation could be obtained by using spiral oscillation and infinite oscillation, as shown in Figure 28. Further analysis of the influence of welding speed on the welding form, they found that when the welding speed of spiral spot was increased from 6 m/min to 18 m/min, the penetration depth of the weld was reduced from 730 μm to 570 μm, and the change remained in the range of 20%. Similar results were obtained for welding with infinite spot shape. And by changing the graphic

frequency from 1 kHz to 1 MHz, the weld width was basically constant. At the interface, the width of the weld began to decrease significantly only when the frequency was increased to 111 kHz, and the penetration depth began to increase significantly when the frequency was lower than 11 kHz. Good weld formation could be obtained after welding regardless of whether spiral or infinite light spots were used.

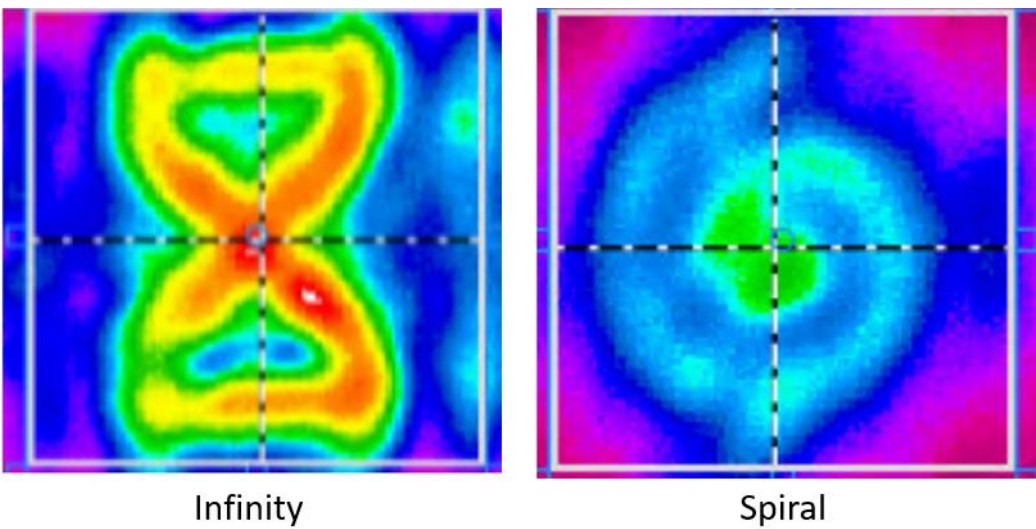

**Figure 27.** Diagram of light spot shape [72].

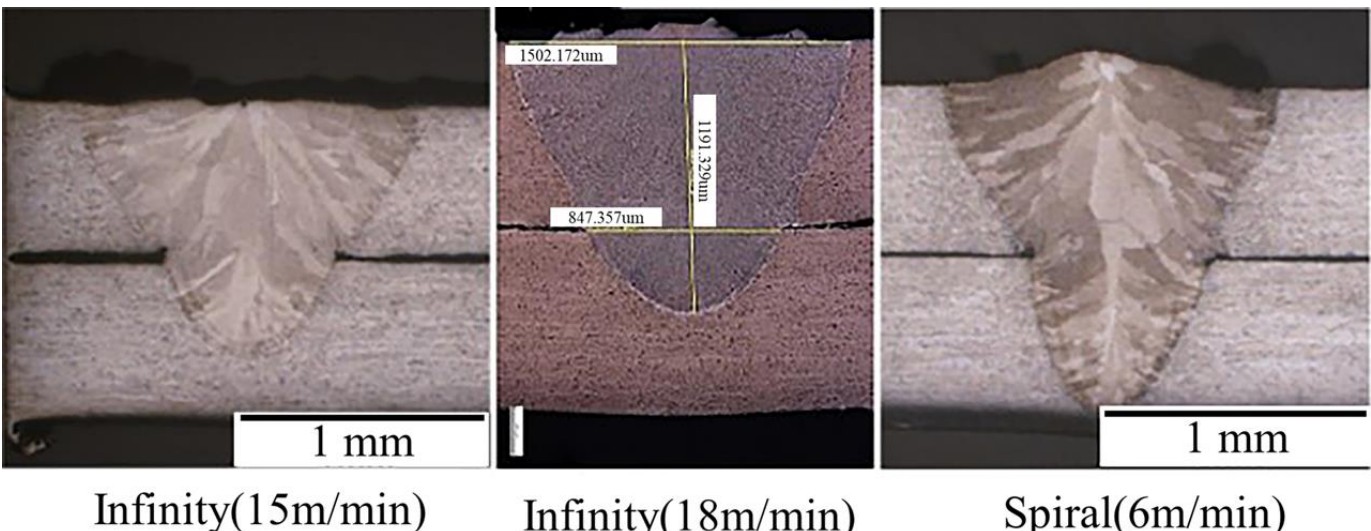

**Figure 28.** Weld forming under different spot shape and speed [72].

*3.2. Welding of Steel Battery Housing*

Stainless steel has been widely used in various applications because of its excellent mechanical properties, corrosion resistance and good welding performance, such as biomedical equipment, nuclear industry components, precision instruments, aviation, automotive industry, batteries and so on [73–78]. The welding of stainless steel usually does not require pre-welding or post-welding heat treatment, because unlike low-carbon steel alloy, there is no Martensite structure in HAZ [79]. Hot cracks are easy to occur during welding of austenitic stainless steel, especially for stainless steel without ferrite [80]. At present, it has been found that the hot crack in stainless steel welding is mainly related to the alloy composition and the content of impurities [81]. Sulfur and phosphorous impurities, in particular, expand the solidification range by forming eutectic liquids with low melt-

ing point [80,82,83]. The Hammar–Svensson chrome–nickel equivalent ratio (Creq/Nieq) during solidification also affects the crack sensitivity [84–86].

Yan et al. [87] used $CO_2$ laser to conduct welding experiments on 5 mm thick 304L stainless steel and successfully obtained good weld joints. They found that no significant cracks or voids were found in the welded joint. The width of the connector was 3.5 mm, and the fusion area of the connector was 6.7 $mm^2$. It could be seen from the microstructure of the joint that the columnar dendrites extended from the fusion boundary to the weld center line, and there was no obvious transition zone and heat-affected zone near the joint. The joint is composed of dark δ-Fe dendrites in austenite matrix, in which δ-Fe existed in a network structure in the fusion zone, and the dendrite spacing was 2–5 μm.

Danny et al. [88] used lap joint welding for stainless steel foil, and they used 60 μm 304 stainless steel foil. They gradually increased the power of the laser from 200 mJ to 400 mJ and the welding speed increased from 0.5 mm/s to 1.5 mm/s. After welding, they found that most of the lap welds showed good surface finish, without the common edge bites and bumps, but with the increase in welding speed and laser pulse energy, the weld roughness increased, accompanied by porosity and collapse. The optimal weld was obtained when the laser pulse energy was 250 mJ, the speed was 0.5 mm/s and the laser pulse frequency was 10 Hz, as shown in Figure 29. Through observation, it could be found that the secondary dendrite distance was submicron level due to the fast cooling speed of laser welding. Further observation by electron microscope showed that the austenite microstructure in the weld had high density dislocation, and there were carbide precipitates in some areas, as shown in Figure 30.

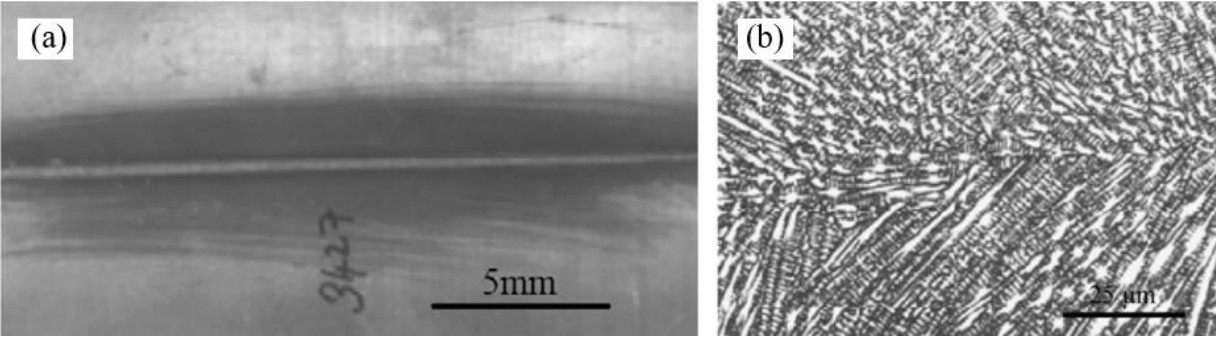

**Figure 29.** Macro morphology of weld (**a**) and microstructure of weld (**b**) (etchant: 10% oxalic acid). Reprinted with permission from Ref. [88]. Copyright 2008, Elsevier.

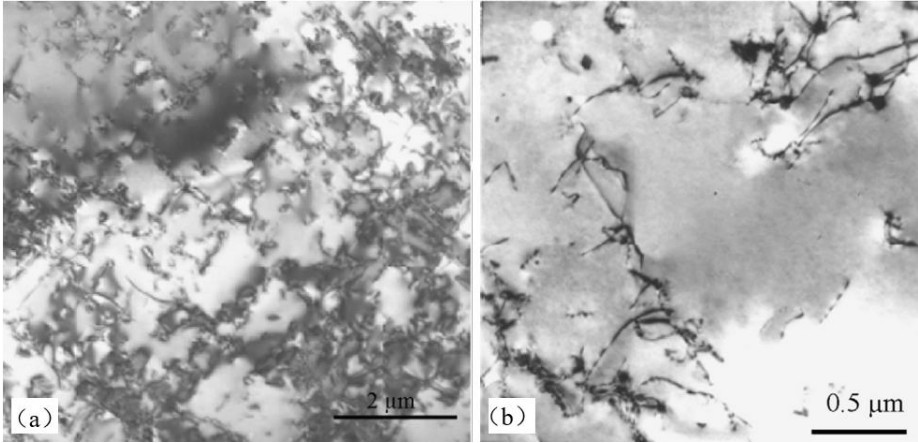

**Figure 30.** Transmission electron micrograph of LBW weld (**a**). Transmission electron micrograph of LBW weld showing the carbide preferentially precipitated at the dislocations (**b**). Reprinted with permission from Ref. [88]. Copyright 2008, Elsevier.

Zhang et al. [89] conducted deep penetration laser welding of 10 mm thick plate stainless steel and studied the influence of welding parameters on weld-formation. When the welding speed increased, the surface shape of the welds gradually became better, the width of the weld gradually became narrower from the width, and the weld nodules at the bottom of the weld also gradually became smaller. Only changing the defocus amount of laser, when the defocus amount was −10 mm, the weld joint forming best. By changing the types of protective gases used in welding process, they found that the maximum weld depth could be obtained by using helium as the protective gas, followed by nitrogen and argon as the shallowest weld penetration. When the laser power could be completely penetrated, no matter what kind of shielding gas was used, the appearance of the weld would not be affected. However, argon gas could be used as shielding gas to protect the bottom of the molten pool, as shown in Figure 31. And the most continuous smooth weld with no spatter and no porosity could be obtained. The weld was examined using XRD, as shown in Figure 32, and it could be seen that the composition of the weld was mainly composed of $\gamma$-Fe and $\delta$-Fe, of which $\gamma$-Fe had a higher content than $\delta$-F. By analyzing the joint of the weld with the optimal welding parameters, as shown in Figure 33a, it could be found that there is no obvious transition zone and heat-affected zone (HAZ) in the weld. The microstructure of the fusion zone was mainly columnar dendrites, which are symmetrically distributed along the weld center. The microstructure of the fusion zone center is equiaxed, as shown in Figure 33b,c. In Figure 33d, the dark dendritic structure was $\delta$-F, where the lighter material was $\gamma$-Fe.

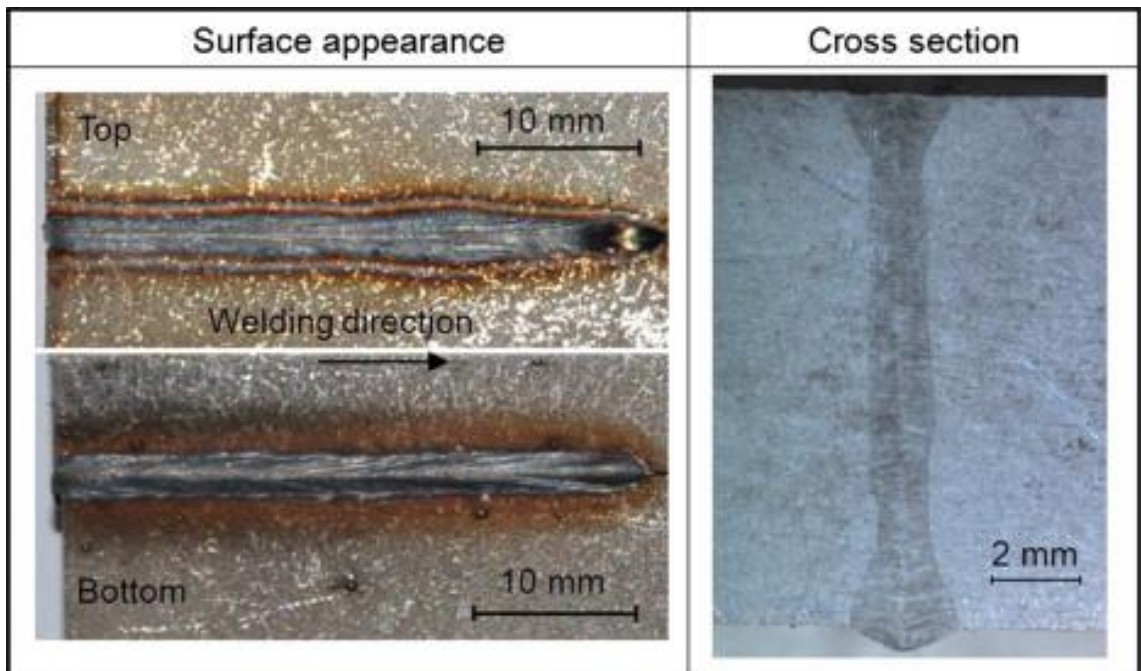

**Figure 31.** Surface appearance and cross-section of optimal butt joint welded at $v$ = 2.4 m/min, $\Delta$ = −10 mm, $q_{top}$ = 30 L/min (N$_2$) and bottom shielding gas flow ($q_{bottom}$) of 5 L/min (Ar). Reprinted with permission from Ref. [89]. Copyright 2014, Elsevier.

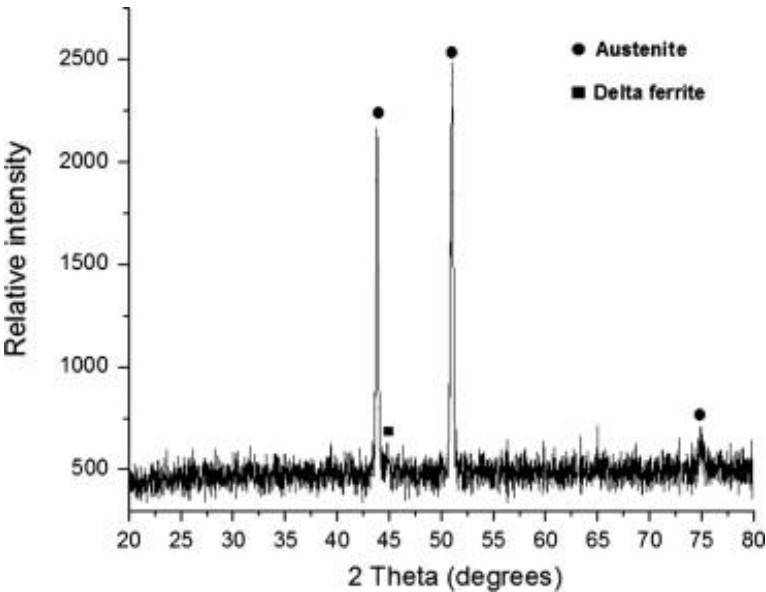

**Figure 32.** XRD patterns of the joint. Reprinted with permission from Ref. [89]. Copyright 2014, Elsevier.

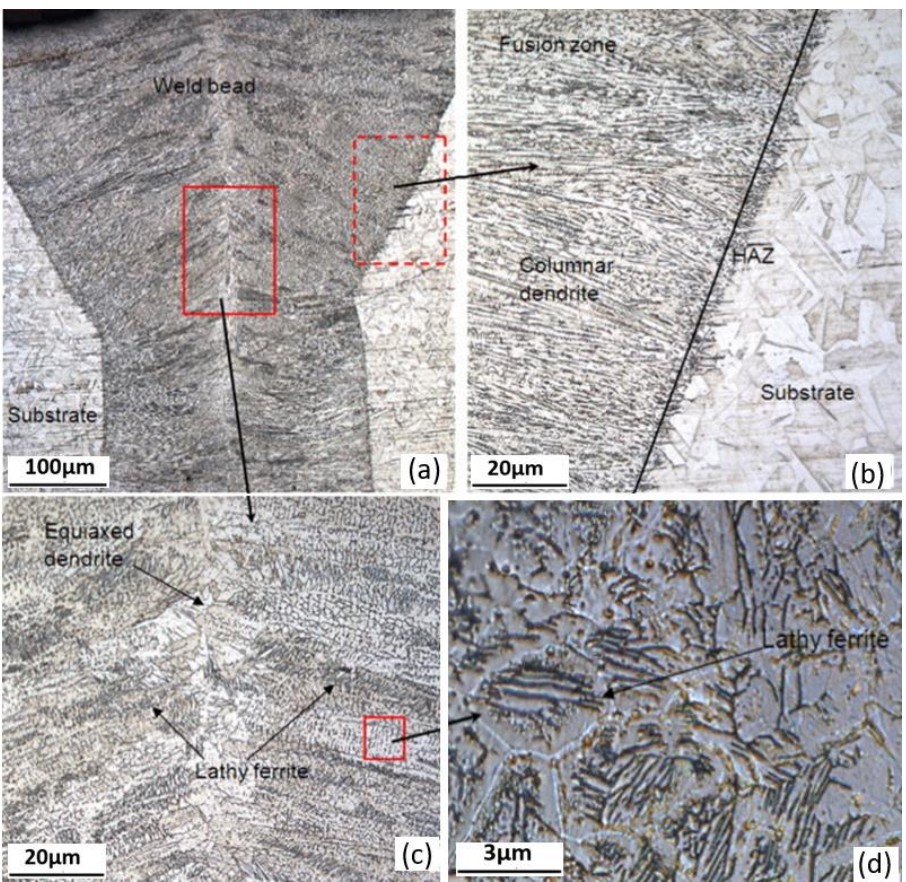

**Figure 33.** Microstructure of optimal butt joint: (**a**) optical micrograph of the entire joint, (**b**) enlarged micrograph between the substrate and fusion zone inside dashed rectangle in (**a**), (**c**) enlarged micrograph of the fusion center zone inside solid rectangle in (**a**), and (**d**) enlarged micrograph of the columnar dendrite zone inside solid square in (**c**). Reprinted with permission from Ref. [89]. Copyright 2014, Elsevier.

## 4. Summary and Outlook

Laser welding is a welding method with high energy density and non-contact and accurate heat input control, which can provide reliable weldability for the welding between dissimilar materials in the battery system of electric vehicles. The laser welding of dissimilar materials has made great progress in the past years. However, no matter the different laser light sources, the optimization of process parameters or the improvement of various joint structures are studied, and there will still be metallurgical defects such as incomplete bonding, brittle metal interphase, corrosion, excessive porosity and cracking. These defects affect the electrical performance and safety of the entire EV battery system. Therefore, if laser welding technology wants to be widely used in the manufacturing of electric vehicle batteries, further research is needed. This paper reviews the research progress of laser welding technology of steel–copper, steel–aluminum, al–copper, Al–aluminum and steel–steel composites is reviewed. Based on the current research, the following suggestions are proposed for future research in this field:

(1) Because the material thickness used in electric vehicle batteries is generally low, it is necessary to optimize the process parameters, accurately control the heat input and improve the welding quality by controlling the thickness of intermetallic compounds. For example, the matching of laser power and welding speed as well as the appropriate beam oscillation frequency.

(2) Appropriate interlayers or coatings are used to change the formation of metal components during welding, so as to adjust the microstructure, improve mechanical properties and reduce resistance.

(3) At present, lasers are still mainly at infrared wavelength (1064 nm), and metals such as copper and aluminum have higher reflectance to the light of the sub-band and have higher absorption rate for blue light (450 nm) and green light (515 nm). Blue light and green light laser can be used for welding experiments.

(4) So far, most of the mechanical properties analyses of laser welded dissimilar material joints have analyzed the static mechanical properties of the joints and few have researched the fatigue properties of the joints. Because electric vehicles are often accompanied by bumps on the road, the performance of these joints under cyclic loads is important. Therefore, it is necessary to carry out more studies on mechanical properties of joints.

**Author Contributions:** Conceptualization: J.F., methodology: J.F. and Z.L., investigation: J.F., P.Z., H.Y., H.S. and Q.L., formal analysis: J.F. and Z.L., writing—original draft: J.F., data curation: P.Z. and D.W., supervision: P.Z., writing—review and editing: P.Z., R.L. and Q.W. visualization: P.Z. and T.S., validation: P.Z., funding acquisition: P.Z. All authors have read and agreed to the published version of the manuscript.

**Funding:** This research was supported by Foundation of Natural Science Foundation of China (52075317 and 51905333), Shanghai Sailing Program (21YF1432300), China Postdoctoral Science Foundation under Grant 2022T150400 and Class III Peak Discipline of Shanghai—Materials Science and Engineering (High-Energy Beam Intelligent Processing and Green Manufacturing).

**Institutional Review Board Statement:** Not applicable.

**Informed Consent Statement:** Not applicable.

**Data Availability Statement:** Data sharing is not applicable to this article.

**Conflicts of Interest:** The authors declare no conflict of interest.

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
