# Peer review of "Application of Laser Welding in Electric Vehicle Battery Manufacturing: A Review"

_coatings, doi:10.3390/coatings13081313_

Round 1

Reviewer 1 Report

This work is a review on application of laser welding in electric vehicle battery manufacturing.

The manuscript initially reports the aim of the work and then review different paper on laser welding between (i) aluminum and steel, (ii) copper and aluminum and (iii) copper and steel. At the end of each paragraph the authors should synthetize in a table the main outcomes/lacks from each paper. These tables should be recalled in the conclusion section to propose new research lines for the welding of such dissimilar joints.

Chapter 3 focuses on laser welding of battery housing. However, the attention to the industrial application should be more stressed. The authors should include pictures of cases studies and discussion of practical challenges. On the other hand, this chapter seems to cover the same aim of chapter 2.

For these reasons, in my opinion the manuscript can be accepted after minor revisions.

Author Response

Reviewer 1

Question 1:

The manuscript initially reports the aim of the work and then review different paper on laser welding between (i) aluminum and steel, (ii) copper and aluminum and (iii) copper and steel. At the end of each paragraph the authors should synthetize in a table the main outcomes/lacks from each paper. These tables should be recalled in the conclusion section to propose new research lines for the welding of such dissimilar joints.

Author response:

I have added a summary and views on the future research direction at the end of each paragraph, and the summary tables are shown in Table 2, Table 4 and Table 6

Question 2:

Chapter 3 focuses on laser welding of battery housing. However, the attention to the industrial application should be more stressed. The authors should include pictures of cases studies and discussion of practical challenges. On the other hand, this chapter seems to cover the same aim of chapter 2.

Author response:

An example is given in Figure 20 of the revised manuscript. The third chapter mainly analyzes the welding of the same material of aluminum and steel, which provides theoretical reference for the actual industrial production.

Reviewer 2 Report

The work is devoted to one of the topical areas of modern technical physics associated with the welding of lithium-ion batteries. The work is review. It discusses the main features associated with the welding of various types of materials used as tires or electrodes in lithium-ion batteries.

I have a few comments about the work:

I) In my opinion, it is necessary to briefly describe and give classical welding methods associated with an electric arc. See, for example, works:

1) DOI 10.1088/1361-6595/ac89a7

2) https://doi.org/10.1016/j.matpr.2022.04.754

II) It should also be noted that when welding different materials, binary alloys can be formed. During their crystallization, defects can be formed, such as, for example, microporosity. See for example the work:

G. Couturier and M. Rappaz, Effect of volatile elements on porosity formation in solidifying alloys, Modell. Simul. Mater. Sci. Eng., 14, 253–271 (2006)

In this regard, the authors need to give methods for dealing with such defects and supplement their review.

III) Figures 16, 17 need to improve the quality

After these remarks are eliminated, the work can be considered for publication in the journal.

English is ok

Author Response

Reviewer 2

Question 1:

In my opinion, it is necessary to briefly describe and give classical welding methods associated with an electric arc. See, for example, works:

1) DOI 10.1088/1361-6595/ac89a7

2) https://doi.org/10.1016/j.matpr.2022.04.754

Author response:

The case of Meng et al. [63] welding 304SS and Cu by laser arc composite welding was mentioned in the manuscript. I found through consulting relevant literature that arc welding alone is not suitable for welding thin foils in battery production. Therefore, no arc welding related content was added.

Question 2:

It should also be noted that when welding different materials, binary alloys can be formed. During their crystallization, defects can be formed, such as, for example, microporosity. See for example the work:

  1. Couturier and M. Rappaz, Effect of volatile elements on porosity formation in solidifying alloys, Modell. Simul. Mater. Sci. Eng., 14, 253–271 (2006)

In this regard, the authors need to give methods for dealing with such defects and supplement their review.

Author response:

At the end of each paragraph of the manuscript, I have added the relevant research success in solving welding defects. For example, Sierra et al. [38], Zhang et al. [39] and Xia et al. [40], proposed solutions for welding aluminum and steel. Solutions proposed by Mei et al. [56] and Weigl et al. [57] for welding copper and aluminum. Solutions proposed by Shen et al. [68] and Shaikh et al. [69] for welding copper and steel.

Question 3:

Figures 16, 17 need to improve the qualityAfter these remarks are eliminated, the work can be considered for publication in the journal.

Author response:

Since the copyright of the pictures cannot be obtained, I have deleted the pictures 12,14,22 in the preliminary manuscript. As for your question about improving picture clarity, I have replaced picture 14 and 15 in the new manuscript with pictures with higher clarity

Reviewer 3 Report

The submitted manuscript represents a very good review of the possibilities to use laser welding in electric vehicle battery manufacturing. The field of battery manufacturing is now constantly growing and a manuscript like this can offer much useful information to readers in order to solve some problems from this field.

I find English translation well as well as graphical illustrations. Also, the number of 84 references cited is appropriate for this type of paper.

I have only two question for the authors and a few technical remarks on the paper:

Question

- Have you maybe considered brazing for joining the parts in battery manufacturing? The temperature in brazing is much lower than welding while the strength of the joints can reach high values. Although, the strength here is not primary. 

- What would be the alternative welding process for applications you analyze? I suggest adding in the Summary section one paragraph on the advantages of laser welding compared to some other process or processes but to mention the alternative.

Technical remarks:

- in the bottom of page 8 the table title should go on page 9,

- similar to Fig. 28 on page 18,

- sections 3.1 and 3.2 have an unequal font of the text.

Author Response

Reviewer 3

Question 1:

 Have you maybe considered brazing for joining the parts in battery manufacturing? The temperature in brazing is much lower than welding while the strength of the joints can reach high values. Although, the strength here is not primary. 

Author response:

The content related to brazing that you mentioned has been included in the manuscript, mainly laser brazing, such as Chen et al. [34], Zhang et al. [39], Xia et al. [40],Hailat et al. [53] and Weigl et al. [57]. They have laser brazing by filling the sandwich.

Question 2:

What would be the alternative welding process for applications you analyze? I suggest adding in the Summary section one paragraph on the advantages of laser welding compared to some other process or processes but to mention the alternative.

Author response:

The new manuscript has introduced the resistance welding and ultrasonic welding commonly used in battery welding, and analyzed the limitations of these two welding methods. At present, the connection of the battery system of new energy vehicles is mainly welding, and a few special cases can also use mechanical connection, which is summarized at the end of the first paragraph of the article.

Reviewer 4 Report

The authors have touched upon a unique topic which is seldom discussed. Though, the topic is unique but i have seen that many of the figures used in the paper are merely cited. I wonder that if copyright has been taken from the respected publisher. 

The takeaway of the surveyed methods is not presented in the paper. I suggest the authors to use a ranking criteria based on the discussed factors so that the reader can have a crisp view point of the methods.  

English language is fine. 

Author Response

Reviewer 4

Question 1:

The authors have touched upon a unique topic which is seldom discussed. Though, the topic is unique but i have seen that many of the figures used in the paper are merely cited. I wonder that if copyright has been taken from the respected publisher. 

Author response:

The copyright problem in the manuscript has been solved, and the parts that cannot be copyrighted have been deleted, such as Figure 12, Figure 14 and Figure 22 in the original manuscript

Question 2:

The takeaway of the surveyed methods is not presented in the paper. I suggest the authors to use a ranking criteria based on the discussed factors so that the reader can have a crisp view point of the methods. 

Author response:

This article mainly investigates the welding defects existing in battery welding and the solutions to these defects, in order to facilitate readers to read. I have included tables at the end of each chapter of the manuscript, which mainly contain the main conclusions of each paper.

Round 2

Reviewer 4 Report

Thank you for addressing my comments

Author Response

Thank you for your valuable suggestions on my manuscript. I have revised the manuscript according to your suggestions. The main purpose of the research is introduced in the new manuscript, and tables are added at the end of each paragraph for readers' convenience. These tables summarize the main conclusions of each paper.